



# Overestimation of closed chamber soil $CO_2$ effluxes at low atmospheric turbulence

Andreas Brændholt[1], Klaus Steenberg Larsen[1,2], Andreas Ibrom[1] and Kim Pilegaard[1]

[1]DTU Environment, Technical University of Denmark, Kgs. Lyngby, 2800, Denmark
[2]Department of Geosciences and Natural Resource Management, University of Copenhagen, Frederiksberg C, 1958, Denmark

*Correspondence to*: Andreas Brændholt (andbr@env.dtu.dk)

**Abstract.**

Soil respiration ($R_s$) is an important component of ecosystem carbon balance and accurate quantification of the diurnal and seasonal variation of $R_s$ is crucial for correct interpretation of the response of $R_s$ to biotic and abiotic factors, as well as for estimating annual soil $CO_2$ efflux rates.

In this study, we measured $R_s$ hourly for one year by automated closed chambers in a temperate Danish beech forest. The data showed a clear diurnal pattern of $R_s$ across all seasons with higher rates during night-time than during day-time. However, further analysis showed a clear negative relationship between flux rates and friction velocity ($u_*$) above the canopy, suggesting that $R_s$ was overestimated at low atmospheric turbulence throughout the year due to non-steady state conditions during measurements. Filtering out data at low $u_*$ values removed or even inverted the observed diurnal pattern, such that the highest effluxes were now observed during day-time, and also led to a substantial decrease in the estimated annual soil $CO_2$ efflux.

By installing fans to produce continuous turbulent mixing of air around the soil chambers, we tested the hypothesis that overestimation of soil $CO_2$ effluxes during low $u_*$ can be eliminated if proper mixing of air is ensured, and indeed the use of fans removed the overestimation of $R_s$ rates during low $u_*$. Artificial turbulent air mixing may thus provide a method to overcome the problems of using closed chamber gas exchange measurement techniques during naturally occurring low atmospheric turbulence conditions. Other possible effects from using fans during soil $CO_2$ efflux measurements are discussed. In conclusion, periods with low atmospheric turbulence may provide a significant source of error in $R_s$ rates estimated by the use of closed chamber techniques and erroneous data must be filtered out to obtain unbiased diurnal patterns, accurate relationships to biotic and abiotic factors, and before estimating $R_s$ fluxes over longer time scales.

# 1 Introduction

Soil respiration ($R_s$) in terrestrial ecosystems is the second largest flux of $CO_2$ after gross photosynthesis and was found to account for 63 % of ecosystem respiration on average in a study of 18 European forest (Janssens et al., 2000; Raich and Schlesinger, 1992). $R_s$ may exhibit both a strong seasonal and diurnal pattern (e.g. Janssens et al., 2000; Tang et al., 2005)



and accurate measurement of $R_s$ at various time scales is thus important to correctly estimate this flux component, i.e. periodic over- or underestimation of $R_s$ may lead to huge errors in annual ecosystem $CO_2$ budget estimates.

$CO_2$ produced in the soil must in the long term be emitted from the soil surface (Maier et al., 2011), although in some ecosystems, leaching of dissolved organic and inorganic carbon may occur (Kindler et al. 2011). $R_s$ is therefore often

measured as soil $CO_2$ efflux by the closed chamber method, which relies on Fick's law of diffusion and steady-state diffusion rate conditions (Gao and Yates, 1998). Under these conditions, the flux rate can be calculated from the increase in chamber $CO_2$ concentration during the chamber deployment period.

To estimate annual $R_s$ on a site, manual chamber measurements are often performed at regular intervals (e.g. twice a month) at multiple plots, which ideally encompass the temporal and spatial variation in $R_s$ in the studied ecosystem. For logistic

reasons manual measurements are, however, most often performed during day-time working hours and therefore do not capture the diurnal variation. To capture the diurnal variation, diurnal measurements campaigns at a few plots are often performed at a high temporal resolution (e.g. every 1-2 hour) but the number of campaigns over a full year may often be limited. Using campaign-wise diurnal cycle patterns for the entire year may cause biases, since the diurnal pattern of $R_s$ may itself exert seasonal differences, which may not be captured during a limited number of diurnal measurement campaigns

(Ruehr et al., 2010).

The assumption behind the usage of closed chambers for estimating soil $CO_2$ efflux may also be challenged and a number of potential biases have been identified that may cause an over- or underestimation of the true soil $CO_2$ efflux (Davidson et al., 2002; Rochette and Hutchinson, 2005; Ryan and Law, 2005). Firstly, the increase in chamber headspace $CO_2$ concentration during measurements decreases the concentration gradient between the soil and chamber headspace, thereby theoretically

decreasing the apparent soil $CO_2$ efflux according to Fick's law of diffusion (Gao and Yates, 1998). Consequently it has often been found that applying a linear fit to the increase in chamber $CO_2$ concentration leads to a systematic underestimation of the measured $CO_2$ efflux compared to the true efflux (Anthony et al. 1995; Venterea, 2010). To correct for this methodological error a non-linear fit can be applied to the increase in chamber headspace $CO_2$ concentration to estimate the $CO_2$ efflux at time zero, where chamber headspace $CO_2$ concentration is at ambient level. Though theoretically sounder, non-

linear fits have been found to increase the uncertainty of the flux estimate (Venterea et al. 2009).

Many chambers are placed on a soil collar permanently installed a few cm into the soil to secure a good sealing of the chamber to the soil surface, thus preventing lateral transport of air into or out from the chamber in the top soil (Healy et al., 1996; Hutchinson and Livingston, 2001; Livingston and Hutchinson, 1995). However, the soil collar may cause a disturbance to the soil e.g. by changing the microenvironment, severing plant roots or even increasing root growth (Görres et

al., 2015). These disturbances of the soil can potentially change the $R_s$ inside the collar compared to undisturbed soil.

Placing the chamber on the soil before a measurement may also lead to a disturbance in the diffusion-driven soil $CO_2$ efflux. This can either lead to a flush of $CO_2$ from the soil pores into the chamber headspace resulting in a higher soil $CO_2$ efflux (Matthias et al., 1980; Hutchinson and Livingston, 1993) or lead to horizontal transport of gas in the soil resulting in a lower soil $CO_2$ efflux (e.g. Conen and Smith, 2000; Kutzbach et al., 2007; Pedersen et al., 2010).



The conditions of the atmosphere surrounding the chamber may also influence the measured $CO_2$ efflux. An over- or under-atmospheric pressure in the chamber headspace can act to either suppress or increase the $CO_2$ efflux respectively (Kanemasu et al. 1974). To maintain ambient pressure in the chamber headspace, closed chambers often have a vent connected to the atmosphere (e.g. Hutchinson and Mosier, 1981; Savage and Davidson, 2003; Xu et al., 2006). High wind speeds moving

over the vent tube may lead to a pressure drop in the chamber headspace, due to the Venturi effect, thereby leading to flux overestimation (Conen and Smith, 1998). Correct vent design has, however, been found to eliminate this effect (Xu et al., 2006).

While the effects mentioned above are well described, the effect of atmospheric turbulence during closed chamber measurements has received much less attention. Recently, a few studies have demonstrated that low friction velocity ($u_*$), a

measure of atmospheric turbulence, can lead to overestimation of fluxes measured by closed chambers, and that it is especially a problem during night-time where $u_*$ typically is lowest (Görres et al., 2015; Lai et al., 2012; Schneider et al., 2009). It has been suggested that the overestimation of chamber fluxes is because a stratified layer of $CO_2$ builds up at the soil surface during periods of low $u_*$, but is broken down by the chamber movement at closure (Görres et al., 2015). The mechanism leading to flux overestimation, however, remains uncertain.

The potential overestimation of soil $CO_2$ effluxes due to low $u_*$ has become relevant especially in recent years when high-frequency soil $CO_2$ efflux measurements have become more widespread by the emergence of commercially available automated closed chamber systems, and increasing integration and usage of chamber flux technologies in international research infrastructures, such as ICOS (Integrated Carbon Observation System), which aims to quantify the greenhouse balance using common approaches and protocols across multiple sites. However, insufficient research has been done on the

topic, and there is no consensus on how to account for this effect to get unbiased measurements of soil $CO_2$ effluxes. Thus it is currently unknown what effect this bias have on up scaled annual soil $CO_2$ efflux estimates for different ecosystems, and how unbiased measurements can be performed during low $u_*$.

Our study had two main aims. The first main aim was to quantify the effect of $u_*$ on closed chamber soil $CO_2$ effluxes in the short-term (i.e. effect on diurnal fluxes) and in the long-term (i.e. effect on annual estimates of $CO_2$ efflux). We measured

soil $CO_2$ effluxes on an hourly basis throughout one year with eight automated closed chambers. In addition we measured soil $CO_2$ effluxes using a manual closed chamber during day-time every two weeks. $u_*$ was measured continuously above the tree canopy. Based on increasing the $u_*$ threshold value in a $u_*$ filtering procedure on the soil $CO_2$ efflux data, we analysed the effect of $u_*$ on the diurnal patterns through the year as well as on the estimate of annual soil $CO_2$ efflux. Secondly, we compared the automatic measurements when filtering at different of $u_*$ threshold values to the manually measured daytime

$CO_2$ effluxes to investigate the consequences of using up-scaled day-time only measurements for estimation of annual soil $CO_2$ effluxes.

The second main aim was to test the hypothesis that the overestimation of soil $CO_2$ effluxes during low $u_*$ was due to insufficient mixing of the air above the soil surface and to test if unbiased soil $CO_2$ efflux measurements could be achieved during low $u_*$ by artificially inducing mixing of the air around the soil chambers. This was done during a 20 day campaign



where fans were used to ensure mixing of the air around the soil chambers at all times for half of the chambers, while the rest of the chambers did not have a fan. This allowed for comparing soil $CO_2$ effluxes measured by chambers with mixing by fans to soil $CO_2$ effluxes measured by chambers that experienced the naturally occurring low atmospheric turbulence.

## 2 Materials and Methods

### 2.1 Site description

Measurements were performed at the Danish ICOS RI site DK-Sor at 40 m a.s.l. (55°29'13' N, 55°38'45' E). Measurements of tower-based eddy-covariance have been running since 1996. The climate is temperate maritime with an annual average temperature of 8.5 °C and an annual average precipitation of 564 mm (Pilegaard et al. 2011).

A dense forest canopy covers the site, and the dominant tree species is European beech (*Fagus sylvatica* L.) with scattered
stands of conifers such as Norway spruce (*Picea abies* (L.) Karst) and larch (*Larix decidua* Mill.) constituting 20 % of the forest (Wu et al. 2013). The stand of beech around the flux tower was planted in 1921 and had an average height of 28 m and an average diameter at breast height of 42 cm in 2010. The understory is poorly developed due to the well-developed canopy, causing a sparsely vegetated forest floor. During spring, however, part of the forest floor is covered by wood anemone (*Anemone nemorosa* L.).

The soil is classified as alfisols or mollisols (depending on the base saturation) with an organic layer with a depth of 10-40 cm. The soil carbon pool (down to a depth of 1 m) is 20 kg m$^{-2}$, with a C/N ratio of about 20 in the upper organic soil layers, dropping to about 10 in the lower mineral layers (Østergård, 2001).

### 2.2 One year campaign of soil $CO_2$ efflux and friction velocity measurements

Soil $CO_2$ efflux was measured automatically over one year from 10 October 2014 to 30 September 2015 (356 days) with five
8100-104 Long-Term $CO_2$ flux chambers and three 8100-101 Long-Term $CO_2$ flux chambers in a multiplexed setup with a LI-8100A Automated Soil $CO_2$ Flux System and a LI-8150 Multiplexer (LI-COR Environmental, Lincoln, Nebraska USA). Measurements were made on permanent, circular soil collars (20 cm diameter) that were inserted 4 cm into the soil prior to the measurement period. The automated chambers each measured soil $CO_2$ effluxes once every hour and were positioned within 15 m of the flux tower.   Soil $CO_2$ efflux was also measured manually at 12 additional circular plots with soil collars
(10 cm diameter) installed 4 cm into the soil as for the automated chambers. The manual plots were positioned close to the automated chamber plots (< 10 meters) and also within 15 m of the flux tower. Soil $CO_2$ efflux at the 12 plots were each measured every two weeks during day-time between 09:00–15:00 CET on days with little or no rain using a portable 8100-102 10 cm survey chamber connected to a LI-8100A Automated Soil $CO_2$ Flux System (LI-COR Environmental, Lincoln, Nebraska USA). Both plots for automated and manual measurements contained bare forest floor including litter but no
shrubs or saplings.



There were three gaps in the automatic data collection. From 12 November to 22 November an unknown system failure occurred. From 3 February to 10 February measurements were stopped to prevent damage to the system due to accumulated snow. From 19 May to 23 July the system did not run due to a mechanical system failure, which required repair. This led to a data coverage of 76 % of the days (272 days out of 356 days), which yielded a total of 52131 individual unique chamber efflux measurements. Chamber closure time was 90 and 150 s for the automated and manual chambers, respectively. Soil temperature and soil moisture content were measured at a depth of 5 cm for both the manual and automated measurements. Alongside the chamber measurements, wind speed in three dimensions was measured by a sonic anemometer (HS-50 Research Anemometer, Gill Instruments Limited, Lymington, UK) at a height of 43 m above soil surface on the flux tower from which $u_*$ was calculated according to Stull (1988).

## 2.3 Fan experiment campaign

During a 20 day campaign in July and August 2016 soil $CO_2$ effluxes were measured at the site with six of the Long-Term $CO_2$ flux chambers, with each chamber measuring soil $CO_2$ effluxes every two hours using a chamber closure time of 5 minutes. The aim of the campaign was to test if artificially increasing the mixing of air around the soil chamber would eliminate the bias of low $u_*$ on measured chamber soil $CO_2$ effluxes. The artificial air mixing for each chamber was provided by 30 cm diameter table fans (Model 546601, HP Schou A/S, Kolding, Denmark) positioned 3 m from the soil chamber and facing the chamber. The fans provided a wind speed of 1.2-1.5 m s$^{-1}$ at the chamber collars. During the first 10 days of the campaign, fans were installed at three chambers, resulting in three chambers with artificial air mixing and three chambers experiencing ambient conditions. During the last 10 days of the campaign the fans were moved to the other three chambers, thus providing a data set with 10 days ambient and 10 days with artificial air mixing on all six chambers. $u_*$ at a height of 43 m above the soil surface was calculated similarly to the one year campaign.

## 2.4 Data analysis

All data analysis was done using R (R Core Team, 2014). Because the current manuscript focuses in the potential error of low turbulent air mixing and because fluxes calculated using non-linear regression fitting may add additional aspects of uncertainty to the calculated fluxes, we calculated the soil $CO_2$ effluxes on a time and area basis by applying linear regression to the increase in chamber $CO_2$ concentration during chamber closure time. An initial period of 20 seconds was discarded after chamber closure (the dead band) for the manual measurements and the one year campaign. For the fan experiment campaign a longer dead band of 60 second was required because an external gas analyser was attached to the LI-8100A. For the automated chamber effluxes, fluxes with an r$^2$ < 0.95 of the linear regression were removed before further analysis, equal to 17 % of the measurements for the one year campaign and 1 % of the measurements for the fan experiment campaign.

The calculated effluxes for the one year campaign were paired with $u_*$ data from the eddy co-variance system in the mast (43 m). The $u_*$ values were used to create sub-datasets by a $u_*$ threshold filtering technique, where effluxes measured at $u_*$ values




lower than a specific threshold value, had been filtered out and removed from the dataset (Aubinet et al. 2000). 12 different $u_*$ threshold values were used, ranging from 0.1 to 1.2 m s$^{-1}$, with a successive higher $u_*$ threshold value of 0.1 m s$^{-1}$. Thus, 12 different sub-datasets each with a specific $u_*$ threshold value were derived from the one year campaign soil $CO_2$ effluxes.

For each of the sub-datasets, diurnal ensemble averages of soil $CO_2$ efflux were calculated for each of the four distinct

seasons at the site. Summer: July and August, autumn: September, October and November, spring: March, April and May and winter: December, January and February.

The annual soil $CO_2$ efflux was obtained for each sub-dataset from the mean soil $CO_2$ efflux for each hour of the day for each month. From this a daily mean was calculated for each month. Monthly soil $CO_2$ effluxes were calculated as the sum of the daily soil $CO_2$ efflux in the respective month and the annual soil $CO_2$ efflux was calculated. One period of data outage

due to system failure (20 May to 22 June) was gap filled by linear interpolation between hourly values of mean diurnal patterns that were calculated from the adjacent periods of the data gap.

In addition to the annual soil $CO_2$ effluxes using all 24 hours throughout the day, the day-time annual soil $CO_2$ efflux was calculated in a similar manner, except that only measurements made between 09:00–15:00 CET were used.

The manually measured soil $CO_2$ effluxes were used to parameterize the empirical model

$$R_s = R_{283} \exp\left[-E_0\left(\frac{1}{T_s + 273.15 - T_0} - \frac{1}{T_s - T_0}\right)\right], \tag{1}$$

of Lloyd and Taylor (1994), where $T_s$ is soil temperature at 5 cm depth and $R_{283}$ is the base respiration at a soil temperature of 10 °C. $T_0$ and $E_0$ are fitted parameters. The model was fitted using nonlinear least squares regression based on a Levenberg–Marquardt algorithm using nlsLM in the R package minpack.lm (Elzhov et al. 2015). The model was used to form a continuous time series of mean daily soil $CO_2$ effluxes throughout the one year by using a continuous measurement of

soil temperature measured at 5 cm depth at the site as input to the model. From the modelled daily soil $CO_2$ effluxes, monthly soil $CO_2$ effluxes were calculated as the sum of the daily soil $CO_2$ effluxes in the respective month. From this the annual soil $CO_2$ efflux was calculated as the sum of all the 12 monthly $CO_2$ effluxes.

## 3 Results

### 3.1 Friction velocity and soil $CO_2$ effluxes

Soil $CO_2$ efflux generally exhibited a diurnal pattern inversely related to the diurnal pattern of $u_*$, with the highest effluxes seen during night-time, when $u_*$ was lowest (Fig. 1). During summer, autumn and spring, $u_*$ showed a clear diurnal pattern with highest values during day-time and lowest values during night-time (Fig. 2a, b, d), while this pattern was weaker during winter (Fig. 2c).

Average hourly soil temperature for the automated soil chambers measured at 5 cm depth showed no diurnality during winter

(Fig. 3c). However, for summer, autumn and spring a slight diurnal pattern was observed with lowest temperatures at 7–10 CET and highest temperatures late in the afternoon or early in the evening (Fig. 3a, b, d).





When comparing the soil $CO_2$ effluxes measured with the automated chambers during the one year campaign with $u_*$, we found a clear relationship with higher soil $CO_2$ effluxes at lower $u_*$ values (Fig. 4), i.e. a significant negative correlation between $u_*$ and soil $CO_2$ efflux for all seasons (P= < 0.001, $r^2$= 0.065). The relationship seemed, however, to level off at a $u_*$ threshold value of approximately 0.7 m s$^{-1}$. Further increasing the $u_*$ threshold value only led to a small or no further

decrease in estimated effluxes.

Based on this result, we then calculated the mean diurnal pattern of soil $CO_2$ efflux for each season during the one year campaign at different $u_*$ threshold values (Fig. 5). With no $u_*$ filtering, the soil $CO_2$ efflux showed a clear diurnal pattern across all seasons with highest effluxes during night-time. This was inversely related to the diurnal pattern of $u_*$ (Fig. 2). The difference between night-time and day-time was most pronounced during summer, where night-time (21–3 CET) effluxes

were 35 % higher than day-time (9–15 CET) effluxes. Applying a successively higher $u_*$ threshold value decreased the difference between day-time and night-time effluxes for all seasons. The most dramatic effect was seen between no $u_*$ threshold value to a value of 0.3 m s$^{-1}$ and 0.5 m s$^{-1}$. Increasing the $u_*$ threshold value from 0.5 m s$^{-1}$ to 0.7 m s$^{-1}$ only led to a slight change in the diurnal pattern. The $u_*$ filtering acted primarily by lowering the high night-time effluxes and only by slightly lowering the day-time effluxes. This uneven lowering of effluxes across the day changed the distinct diurnal pattern

with a large difference between night-time and day-time effluxes, i.e. at a $u_*$ threshold value of 0.5 m s$^{-1}$ and 0.7 m s$^{-1}$, the diurnal pattern of soil $CO_2$ efflux for summer was more uniform across the day, with only slightly lower effluxes in the afternoon. The change in the diurnal pattern in response to an increased $u_*$ threshold value was also seen for the other seasons. For winter and spring, the diurnal pattern of soil $CO_2$ efflux became more uniform, with no apparent difference between night-time and day-time fluxes (Fig. 5k to 5o and 5p to 5t, respectively). For autumn, applying a $u_*$ filtering

procedure reversed the diurnal pattern from the highest effluxes being seen during night-time, to the highest effluxes being seem during day-time.

Looking at mean daily soil $CO_2$ effluxes from the automated chambers revealed that they generally followed the soil temperature throughout the year (Fig. 6a). The same was found for the soil $CO_2$ effluxes based on the manual chamber measurements and the output of the empirical model (Fig. 6c). A high day to day variability was found throughout the year

for the automated measurements, which was especially pronounced during summer. Although following the same seasonal pattern, the modelled effluxes based on the manual measurements were slightly lower than those from the automated measurements (summer: 2.97 for the manual vs. 4.10 µmol m$^{-2}$ s$^{-1}$ for the automated measurements,  autumn: 2.36 for the manual vs. 2.86 µmol m$^{-2}$ s$^{-1}$ for the automated measurements, winter: 0.485 for the manual vs. 0.608 µmol m$^{-2}$ s$^{-1}$ for the automated measurements) except for spring where the manual measurements were the highest (1.19 for the manual vs. 1.04

µmol m$^{-2}$ s$^{-1}$ for the automated measurements). Applying a successively higher $u_*$ threshold value to the automated soil $CO_2$ effluxes generally decreased the daily mean soil $CO_2$ effluxes, as well as the day to day variability (Fig. 6b). The biggest decrease was seen between no $u_*$ filter to a $u_*$ threshold value of 0.7 m s$^{-1}$. Further increasing the $u_*$ threshold value only led to minor additional decreases in the mean daily soil $CO_2$ effluxes. At a $u_*$ threshold value of 0.7 m s$^{-1}$ the mean daily soil



$CO_2$ effluxes for summer, autumn, winter and spring was lowered with 25 % (to 3.09 µmol m$^{-2}$ s$^{-1}$), 17 % (to 2.37 µmol m$^{-2}$ s$^{-1}$), 19 % (to 0.492 µmol m$^{-2}$ s$^{-1}$) and 18 % (to 0.856 µmol m$^{-2}$ s$^{-1}$) respectively in comparison to when no $u_*$ filter was applied.

### 3.2 Annual soil $CO_2$ efflux

Annual soil $CO_2$ effluxes, based on the automated chamber flux data for the one year campaign, were calculated with the exclusion of data at different $u_*$ threshold values (Fig. 7). The highest annual soil $CO_2$ efflux was found when no $u_*$ filter was used (808.9 g C m$^{-2}$ yr$^{-1}$). Increasing the $u_*$ threshold value decreased the annual soil $CO_2$ efflux, with the largest decrease of 21 % observed between unfiltered data to a $u_*$ threshold value of 0.7 m s$^{-1}$ (from 808.9 to 641.7 g C m$^{-2}$ yr$^{-1}$). Between a $u_*$ threshold value of 0.7 to 1.2 m s$^{-1}$, only a small decrease in annual soil $CO_2$ efflux of 7 % was seen (from 641.7 to 596.9 g C

m$^{-2}$ yr$^{-1}$). The annual soil $CO_2$ efflux from the empirical model based on the manual chamber measurements was 666.6 g C m$^{-2}$ yr$^{-1}$.

### 3.3 Day-time soil $CO_2$ effluxes vs. daily effluxes

To assess the consequences of using only day-time soil $CO_2$ efflux data, instead of data for the entire day, when upscaling to annual soil $CO_2$ efflux, we calculated annual soil $CO_2$ effluxes for the one year campaign at different $u_*$ threshold values

using only fluxes measured between 9–15 CET, and compared these to the annual soil $CO_2$ effluxes calculated using effluxes for the entire day at different $u_*$ threshold values (see section 3.2).

At no $u_*$ filter, the annual day-time soil $CO_2$ efflux was 13 % lower than the annual entire day soil $CO_2$ efflux (808.9 vs. 703.3 g C m$^{-2}$ yr$^{-1}$ for the annual entire day and day-time soil $CO_2$ effluxes respectively. Fig. 8). Increasing the $u_*$ threshold value decreased the difference between annual day-time soil $CO_2$ effluxes and entire day effluxes, with a steep decrease from

13 to 4.5 % observed between no $u_*$ filtering to a $u_*$ threshold value of 0.3 m s$^{-1}$ (705.4 vs. 673.7 g C m$^{-2}$ yr$^{-1}$ for the annual entire day and day-time soil $CO_2$ effluxes respectively). Further increasing the $u_*$ threshold value only resulted in a minor change in the relationship between the annual day-time soil $CO_2$ efflux and the entire day efflux. However, they were almost identical at a $u_*$ threshold value of 1.2 m s$^{-1}$ (596.9 vs. 595.5 g C m$^{-2}$ yr$^{-1}$ for the annual entire day and day-time soil $CO_2$ effluxes respectively).

### 3.4 Fan experiment campaign

The observation that $u_*$ filtering profoundly affected estimated soil $CO_2$ effluxes led us to design a simple fan experiment to test if the bias of low atmospheric turbulent mixing on the measured chamber soil $CO_2$ effluxes could be eliminated by ensuring adequate mixing of air around the soil chamber. When no fan was installed, a significant negative relationship was found between soil $CO_2$ efflux and $u_*$ (r$^2$ = 0.040, P = < 0.001, slope = -0.377, data now shown) comparable to the one year

campaign (Fig. 4). However, with fans installed the negative relationship changed into a significant positive relationship (r$^2$




= 0.080, P = < 0.001, slope = 0.353), clearly indicating a strong effect of installing fans. With no fans, the soil $CO_2$ efflux showed a clear diurnal pattern with highest effluxes during night-time (Fig. 9a and 9b). However with fans, the opposite diurnal pattern with highest effluxes during day-time was seen. The change in diurnal pattern when using a fan was primarily due to a decrease of the high night-time effluxes (50 % lower for effluxes measured at 21–03 CET). However, a decrease in

day-time effluxes was also observed when using the fans (26 % lower for effluxes measured at 9–15 CET).

## 4 Discussion

### 4.1 Friction velocity and soil $CO_2$ effluxes

Soil $CO_2$ effluxes generally followed the seasonal changes in soil temperature, being highest in summer and lowest in winter. Our one year measurement series with a total of 52131 individual soil $CO_2$ efflux measurements from 8 automatic chambers

clearly showed a negative relationship with the level of atmospheric turbulence ($u_*$), which was especially visible during periods with similar soil temperature, where apparent rates of soil $CO_2$ effluxes were clearly higher at lower compared to higher $u_*$ values.

For closed chamber measurements to represent real $R_s$, we assume steady-state diffusion from the source in the soil to the atmosphere according to Fick's law. Accordingly, there must be a constant concentration of atmospheric $CO_2$, for the flux to

be stable, and the physical application of a closed chamber should not change or break down any $CO_2$ gradients. Otherwise advection may take place. Our data strongly indicate that these assumptions are likely not met during low $u_*$, when the high soil $CO_2$ effluxes were observed.

$u_*$ typically varies on a daily basis with low $u_*$ during calm nights and high $u_*$ during daytime (Stull, 1988). This is in agreement with our results (Fig. 2). The low atmospheric turbulence during calm nights has been found to cause a build-up

of $CO_2$ above the soil, because of improper mixing of the layer of air above the surface (Brooks et al., 1997). This lowers the concentration gradient of $CO_2$ from the soil to the atmosphere causing a build-up of $CO_2$ in the soil (Wohlfahrt et al., 2005; Flechard et al., 2007). This acts to supress the apparent soil $CO_2$ efflux, and may also lead to a higher apparent soil $CO_2$ efflux once turbulence is re-established, because of the subsequent release of the $CO_2$ that has been build up in the soil (Massman et al., 1997). Our chamber measurements, however, show higher and not lower soil $CO_2$ effluxes during low $u_*$. It

is unlikely that $u_*$ has a direct effect on the biological activity of bacteria, fungi and/or plant roots in the soil and thus it seems evident that the apparent increase in soil $CO_2$ efflux measured during low $u_*$ is a measurement bias of the closed chamber technique. Only a few studies have demonstrated the potential problem of using closed chambers during periods with low atmospheric turbulence (Görres et al., 2016; Koskinen et al., 2014; Lai et al., 2012; Schneider et al., 2009). Görres et al. (2016) suggested that during low atmospheric turbulence, the chamber causes a disturbance and/or mixing of the

stratified layer of still air above the soil surface, as the chamber moves on to the soil at the beginning of a measurement, while Lai et al. (2012) accredited the error to be caused by a similar effect by the internal chamber fan. In both cases, the $CO_2$ rich air just above the soil surface is instantly mixed with less $CO_2$ rich air from above. This causes a sudden drop in the





$CO_2$ concentration just above the soil surface, which in turn increases the concentration gradient between the soil and the atmosphere, thus leading to an apparent high $CO_2$ efflux being measured by the chamber (Görres et al., 2016). Our results support this hypothesis. In addition, we observed that soil temperatures were similar in the data sets with different $u_*$ threshold values (data not shown), indicating that the relationship between flux rates and $u_*$ were not confounded by such

potential differences in the different subsets of observations. We also showed that this error of overestimation of soil $CO_2$ effluxes during low $u_*$ had a dramatic effect on both the diurnal pattern of soil $CO_2$ efflux, and on the estimate of annual soil $CO_2$ efflux, as we will discuss this in the following sections.

## 4.2 Diurnal pattern of soil $CO_2$ effluxes

The hourly measurements of soil $CO_2$ effluxes showed a clear diurnal pattern, with generally highest effluxes during night-
time, when no $u_*$ filter was applied to the data (Fig. 5a, 5f, 5k, 5p). Soil temperature at 5 cm depth, however, did not show a diurnal pattern for winter, and only a slight diurnal pattern for the other seasons with highest temperatures late in the afternoon or early in the evening (Fig. 3). This could indicate a hysteresis between soil $CO_2$ efflux and soil temperature for summer, autumn and spring. A few studies with automated closed chambers have found a similar hysteresis between soil $CO_2$ efflux and soil temperature (e.g. Phillips et al., 2011; Savage et al., 2013; Tang and Baldocchi, 2005). The hysteresis has
been explained as a result of priming of the soil bacteria by carbon exudates from plant roots (Kuzyakov and Gavrichkova, 2010). During day-time in the growing season, plants assimilate carbon via photosynthesis. Part of the assimilated carbon go to the roots via the phloem, and is released into the rhizosphere (Kuzyakov, 2002). Once in the soil, this carbon is readily consumed by bacteria leading to an increase in $R_s$. Even though photosynthesis takes place during day-time, the increase in $R_s$ has been found to lag after photosynthesis, because the photo assimilates need to be transported from the leaves to the
roots via the phloem. Varying lag times have been found, with the shortest times for short plants such as grasses, and lag times of up to 4–5 days for mature trees (Kuzyakov and Gavrichkova, 2010). However, when applying a successively higher $u_*$ threshold value, the diurnal patterns of soil $CO_2$ efflux across all seasons changed, mostly due to removal of the overestimated night-time effluxes. For our measurements, the overestimation of chamber effluxes due to low $u_*$ thus works as a selective systematic error that mostly applies to night-time of the diurnal pattern. This is in agreement with other studies
(Görres et al., 2016; Koskinen et al., 2014; Lai et al., 2012; Schneider et al., 2009). During summer, we observed that the difference between night-time and day-time was smaller than during spring and autumn, and only the fluxes between 10–20 CET were slightly lower than the night-time fluxes. It is possible that this pattern is due to an increase in the night-time soil respiration due to priming of the soil organisms via carbon flow from the roots into the rhizosphere and soil. This was, however, not tested in the present study. For the other seasons priming is not as likely, since only low or no photosynthesis
takes place during this time. During winter and spring, the diurnal patterns changed from a clear diurnal pattern with highest night-time fluxes to no difference between night and day-time fluxes when applying the $u_*$ filtering. During autumn, it changed from a clear diurnal pattern with highest night-time fluxes, to a pattern where the fluxes seem to be slightly higher at daytime.



Our results show that overestimation of soil $CO_2$ effluxes at low $u_*$ can change the apparent diurnal pattern of soil $CO_2$ effluxes, and they highlight the importance of taking this source of error into account, since negligence of the problem may lead to misinterpretation of the relationship between $R_s$ and its physical drivers like temperature and soil humidity, as well as lead to erroneous estimation of lag times between $R_s$ rates and flow of carbon from recent plant assimilates. Much research

has been done on time lags between inputs of carbon via photosynthesis and $R_s$ using closed chambers (Högberg et al., 2001; Tang et al., 2005). We expect that it is possible that overestimates of soil $CO_2$ effluxes at low $u_*$ may have influenced the interpretation of the diurnal pattern in some previous studies that have not taken overestimation of effluxes during low $u_*$ into account. It could therefore be valuable to re-evaluate these studies by including data of $u_*$ if such data are available.

### 4.3 Annual soil $CO_2$ effluxes

The $u_*$ filtering had a considerable effect on the annual soil $CO_2$ efflux estimates based on the automated chamber measurements by decreasing the annual efflux in response to increasing the $u_*$ threshold value. This could be expected from the observed negative relationship between soil $CO_2$ efflux and $u_*$. The annual soil $CO_2$ efflux modelled from the manual day-time-only chamber measurements measured at 12 soil collars within 10 meters from the automated chambers was 666.6 g C m$^{-2}$ yr$^{-1}$. During day-time (9–15 CET) $u_*$ was usually high (Fig. 2). The risk of overestimation of a fraction of the manual

soil $CO_2$ efflux measurements due to low $u_*$ can therefore be viewed as minor. The automated chamber annual soil $CO_2$ efflux including all data was 808.9 g C m$^{-2}$ yr$^{-1}$, i.e. a value much higher than the 666.6 g C m$^{-2}$ yr$^{-1}$ from the manual chambers. However, the annual soil $CO_2$ efflux estimate from the automated chamber measurements decreased to 663.4 g C m$^{-2}$ yr$^{-1}$ and 596.8 g C m$^{-2}$ yr$^{-1}$ at a $u_*$ threshold value of 0.5 m s$^{-1}$ and 1.2 m s$^{-1}$, respectively, i.e. much closer to the annual soil $CO_2$ efflux from the manual measurements (Fig. 7). Thus, it seems that the automated measurement and the manual day-

time measurements provide comparable estimates of annual soil $CO_2$ effluxes, when the effect of low $u_*$ is accounted for, as part of the quality checking procedure for the automated measurements. This seems to increases the confidence in the much less frequent manual measurements, and potentially shows that day-time only measurements, at least in our case, was not a major source of error for the upscaling to an annual estimate of soil $CO_2$ efflux. To exemplify this, we compared the annual soil $CO_2$ effluxes from the automated chambers with estimates of annual soil $CO_2$ effluxes based on only day-time (9–15

CET) soil $CO_2$ effluxes from the automated chambers. The day-time annual soil $CO_2$ efflux was 13 % lower than the annual entire day soil $CO_2$ efflux with no $u_*$ filter, and decreased to 4.5 % at a $u_*$ filter of 0.3 m s$^{-1}$ (Fig. 8) i.e. resulting in more comparable annual soil $CO_2$ effluxes when the overestimation of $u_*$ is accounted for.

### 4.4 Fan experiment

Mixing of the air around the soil chambers with fans had a considerable effect by decreasing the measured soil $CO_2$ effluxes

(Fig. 9). The biggest difference was seen during night-time, but even at day-time a smaller effect was observed. This selective lowering of effluxes changed the apparent diurnal pattern, such that the highest fluxes now were measured during




day-time. By using a fan we hypothesised that the effect of breaking the stratified layer by the chamber during low $u_*$ would not occur because the artificially induced wind by the fan continuously ensured proper mixing of the air around the chamber and thus likely prevented a build-up of a stratified layer of $CO_2$. We believe that the assumption for steady state rate of diffusion of $CO_2$ out of the soil is closer to being fulfilled with a fan, since the proposed mechanism for the apparent higher

soil $CO_2$ efflux at low $u_*$ can no longer take place, due to mixing of the air by the fan prior to the measurement. Thus chamber soil $CO_2$ measurements are no longer overestimated at low $u_*$, which is also seen by that there was no negative relationship between soil $CO_2$ effluxes and $u_*$ when a fan was used. However, when using the fans we also observed an apparent decrease in the measured day-time soil $CO_2$ effluxes, when $u_*$ was generally high. We suggest three potential causes of this difference in the day-time effluxes. Firstly, the fan experiment campaign ran over 20 days, i.e. 10 of the days

were with a fan installed and 10 days were without a fan. Thus a comparison for a chamber is made at different times, and differences in environmental conditions may have caused differences in the rates of $R_s$ between the two periods. Indeed, a difference in the mean soil temperature at 5 cm depth of 1 °C was found between the two periods. However, the difference in soil $CO_2$ effluxes was found for both groups of chambers, i.e. both for the ones with a fan installed during the cold period and for those with a fan installed during the warm period, and thus the potential directional bias of differences in soil

temperature during the two periods can be ruled out. Secondly, the slight decrease of day-time effluxes when fans were installed could be because low $u_*$ was sometimes observed also during day-time, although to a much lesser degree than during night-time. A similar small decrease of the day-time effluxes for the one year campaign when increasing the $u_*$ threshold value was also observed (Fig. 5), highlighting that there may still be such an effect even during day-time. Thirdly, it is possible that although a fan eliminated the bias of low $u_*$ on chamber effluxes, the wind induced by the fan introduced

another potential bias to the measurement. Steady-state diffusion rate of $CO_2$ out of the soil with and without a chamber is a requirement for unbiased chamber measurements of soil $CO_2$ efflux. However, the rate of diffusion of a gas out of the soil is known to be highly sensitive to wind speed at the soil surface, and higher wind speeds can increase the diffusion rate or even cause advective transport out of the soil (Janssens et al., 2000; Roland et al., 2015). This effect is well-known to cause severe overestimation of soil $CO_2$ effluxes when soil chambers are equipped with a heavy internal fan because the resulting wind

speed inside the chamber is much higher than outside (e.g. Hanson et al., 1993; Hooper et al., 2002; Le Dantec et al., 1999). The LI-8100A chambers used in our experiments do not have an internal fan. Instead they rely on the air movement that is created when air is pumped to and from the chamber in combination with the spherical shape of the chamber for adequate mixing of chamber air (LI-COR Biosciences, 2015). The wind speed in the LI-8100A chamber may therefore be fairly low compared to outside conditions. It is therefore possible that we see the opposite effect of what is seen in chambers with an

internal fan, namely that the wind speed in the chamber is lower than outside, resulting in a lower rate of diffusion and consequently a lower measured soil $CO_2$ efflux, and that this effect increases with the installation of fans that increase the wind speed outside the chamber. A similar effect has also been found to occur during natural conditions outside the chamber where soil effluxes measured by micrometeorological methods have been found to increase with increasing ambient wind speed (Denmead and Reicosky, 2003). To eliminate this potential bias due to difference in wind speed, we suggest a closer





matching of the chamber wind speed with the ambient wind speed. It is possible that a lower wind speed induced by a fan would be adequate to eliminate the effect of low $u_*$. This would potentially lower the difference in wind between the outside and inside of the chamber, thus minimizing the potential bias due to different wind speeds. This was, however, not tested in the current study.

**4.5 Towards unbiased soil $CO_2$ effluxes**

Chamber-based measurements of soil efflux of $CO_2$ and other greenhouse gases play an important role in constraining the global carbon cycle. Thus it is crucial to obtain unbiased measurements that are as close to the true rates of emissions or uptake as possible. Many biases have already been quantified or even eliminated through methodological and technical advances. Here we showed that the bias of soil $CO_2$ efflux measurements at low $u_*$ can be accounted for by using a $u_*$

filtering procedure. However, we do see a number of challenges with this methodology that needs to be addressed. One challenge is to choose the correct $u_*$ threshold value. A similar challenge has faced the eddy covariance method, where multiple methods have been developed to determine the right $u_*$ threshold value (Gu et al., 2005). One method is to subjectively choose the appropriate level e.g. based on visual inspection of scatter plots of night-time fluxes versus $u_*$ (Gu et al., 2005). For our data, this would mean choosing a $u_*$ threshold value, where further increasing the $u_*$ threshold value would no longer change the mean soil $CO_2$ effluxes observed. When looking at the diurnal patterns in Fig. 5, only a small

further change in the diurnal pattern is seen between a $u_*$ threshold value of 0.5 to 0.7 m s$^{-1}$, thus indicating that a $u_*$ threshold value between 0.5 and 0.7 m s$^{-1}$ would be appropriate. A slightly higher $u_*$ threshold value seems to apply for the mean hourly soil $CO_2$ effluxes aggregated into groups based on $u_*$ values (Fig. 4), where no decrease in effluxes is seen after the binned group with $u_*$ values of 0.7–0.8 m s$^{-1}$. Interestingly, a similar levelling off of the decrease in the estimated annual soil $CO_2$ efflux in response to an increasing $u_*$ threshold value was not seen (Fig. 7). Instead, the annual soil $CO_2$ efflux

continued to decrease slightly even after the big decrease, that was seen up to a $u_*$ threshold value of 0.7 m s$^{-1}$. This indicates that effluxes are still to some extend influenced by $u_*$ at these high levels. We analysed the data to see if the $u_*$ filtering had an influence on soil temperature, which potentially could selectively remove higher effluxes measured at higher temperatures, leaving only measurements in colder periods at high $u_*$ threshold values (data not shown). The results could, however, not explain the continuous decrease in soil $CO_2$ efflux. Thus it remains an open question why this decrease is still

seen.

Another method to determine the $u_*$ threshold value is to use objective statistical methods such as the Moving Point Test also commonly used in eddy covariance data analysis (Gu et al., 2005). We suggest testing some of these methods for determination of the $u_*$ threshold value for chamber $CO_2$ effluxes in future studies.

We tried to see if using a stricter flagging of data based on the r$^2$ value of the linear fit to the increase in $CO_2$ concentration during chamber closure could also be used to remove overestimated effluxes (data not shown). A similar effect of $u_*$ on effluxes could, however, still be seen even when a higher r$^2$ value was used to quality check the data, indicating that even





during low atmospheric turbulence the increase in chamber $CO_2$ concentration often followed the expected pattern. High $r^2$ values and near-linear increases in chamber $CO_2$ concentration during chamber deployment alone can therefore not be used as the only means of quality checking, as this may still lead to falsely accepting overestimated measurements during low turbulence.

Although we have here shown that a $u_*$ filtering procedure seems to have a good potential to be used for quality control of closed chamber measurements of soil $CO_2$ efflux, it ultimately leads to data gaps that need to be filled using the appropriate gap filling techniques e.g. based on established empirical relationships between measurement bias and $u_*$ (Lai et al., 2012). Data gaps, however, equal loss of data and thus methods should be developed to be able to observe unbiased chamber measurements also during periods with low atmospheric turbulence. Valid chamber measurements during periods with low

$u_*$ are especially important since this coincide with periods of data gaps in fluxes estimated by the eddy covariance technique. Thus it is clear that chamber measurements during these periods can't be used as ground truth for the eddy covariance flux estimates (Görres et al., 2016). Our results from the fan experiment have shown promising results in terms of removing the negative relationship between $u_*$ and soil $CO_2$ effluxes. This is, to the best of our knowledge, the first time this has been attempted, and we expect that this method has a future potential. A few others have touched upon how to get

unbiased chamber effluxes during low $u_*$. Lai et al. (2012) found that closed chamber $CH_4$ and $CO_2$ effluxes were overestimated at low $u_*$. However, by increasing the chamber closure time to 30 minutes, they found that fluxes were no longer influenced by $u_*$ after 13 minutes of chamber closure time, thus yielding reliable fluxes. However, it can be argued that longer chamber closure times introduce other potential biases for most closed chamber systems, because the chamber headspace concentration of $CO_2$ or $CH_4$ changes more and therefore affects the concentration gradient between soil and

atmosphere, which ultimately can lead to underestimation of efflux rates instead. We only used a chamber closure time of 90 seconds for the one year campaign, because this time was sufficient to adequately measure an increase in chamber headspace $CO_2$ concentration over time. We did, however, test if flux estimates were affected by using different dead bands of 10, 20, 30 and 40 seconds, respectively, or by using linear or non-linear flux calculation methods (data not shown). In all cases we observed a similar pattern of overestimation of soil $CO_2$ effluxes during low $u_*$ as reported here for linearly calculated

effluxes with a dead-band of 20 s. Thus neither dead band nor flux calculation method could eliminate the effect of low $u_*$ on soil $CO_2$ effluxes.

Even though automated closed chamber systems are based on the same principle of passive, steady-state diffusion of gases between the soil and the chamber headspace during chamber closure, they come in a wide variety of designs with various shapes and sizes. We expect that chamber design may influence the effect of $u_*$, although it remains to be tested. Görres et al.

(2016), however, also argued that automated chambers have the potential to provide unbiased chamber fluxes during low atmospheric turbulence by fulfilling certain design criteria that ensure that the stable atmospheric layer of air above the soil surface is not broken up during a measurement. These design criteria include a low chamber height of less than 20 cm and a low chamber closing speed in the horizontal plane, both aiming at keeping the chamber in the same horizontal plane, where there is no steep $CO_2$ gradient. This approach is directly in contrast with the approach we used with fans, where we ensured

mixing of the air above the soil surface at all times, such that any stratification was eliminated before the beginning of a measurement. The proposed design criteria may lead to a lower disturbance of the air column above the soil surface, but it is uncertain if it is possible to eliminate disturbance of air completely. One of the fundamental principles of closed chamber measurements is mixing of air inside the chamber. Even with a low chamber height we suspect that mixing of stratified air

just few cm from the soil surface may lead to an overestimated flux during low $u_*$.

## 5 Conclusions

The study had two major results. Firstly, we showed that soil $CO_2$ effluxes measured by automated closed chambers were overestimated at low $u_*$ throughout one year in a Danish beech forest and that this overestimation considerably biased the diurnal patterns of soil $CO_2$ effluxes and led to an overestimation of the annual soil $CO_2$ efflux. We recommend that any

analysis of soil $CO_2$ efflux measured by automated closed chambers must consider overestimation of effluxes at low atmospheric turbulence, to yield unbiased estimates of soil $CO_2$ effluxes. This is crucial when investigating temperature responses of $R_s$ and biological links in ecosystems between $CO_2$ production and $R_s$ on a short time scale, but also for correct estimation of annual soil $CO_2$ effluxes. We have shown that a $u_*$ filtering procedure can be used to remove data that are influenced by insufficient turbulence. The drawback of this post-processing method is, however, a loss of data and thus a

loss of information during atmospheric conditions where the eddy covariance method cannot be applied. Our analysis highlights the need for methodological developments, which will allow for unbiased chamber measurements to be made also during low atmospheric turbulence. We see the results from our fan experiments as a significant step along the way. Here we showed that ensuring continuous mixing of air around the soil chamber by a fan eliminated the overestimation of soil $CO_2$ effluxes due to low $u_*$, thus enabling reliable chamber measurements even at low $u_*$, even though the continuous mixing of

air may have introduced a new bias during chamber measurements. Additional studies are needed to further explore this approach.

## Acknowledgements

This study was funded by the free Danish Ministry for Research, Innovation and higher Education, the free Danish Research Council (DFF – 1323-00182).

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




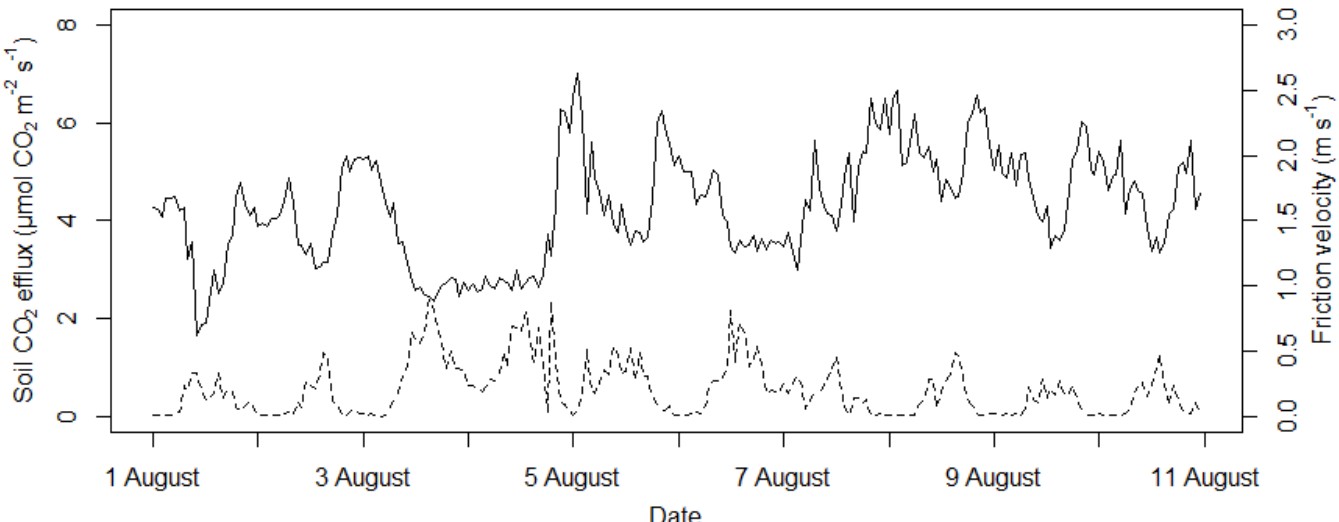

**Figure 1: Example of mean hourly soil $CO_2$ effluxes from the eight automated chambers (solid line) and mean hourly friction velocity ($u_*$) at 43 m above the soil surface (dashed line) for 10 days during August 2015.**

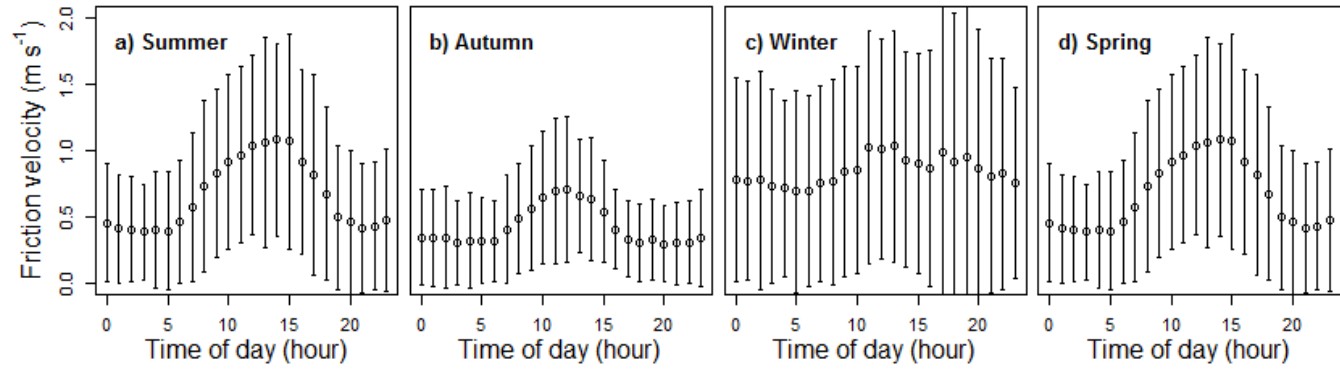

**Figure 2: Mean (± standard deviation) diurnal pattern of friction velocity ($u_*$) at 43 m above the soil surface for summer (a), autumn (b), winter (c) and spring (d).**



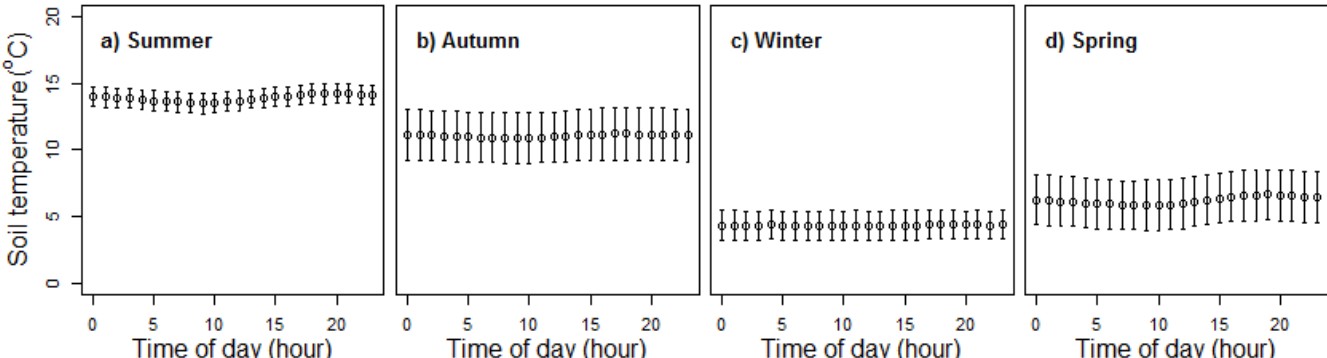

**Figure 3: Seasonally averaged diurnal pattern of soil temperature (± standard deviation) at 5 cm depth measured at the eight automated chambers for summer (a), autumn (b), winter (c) and spring (d).**

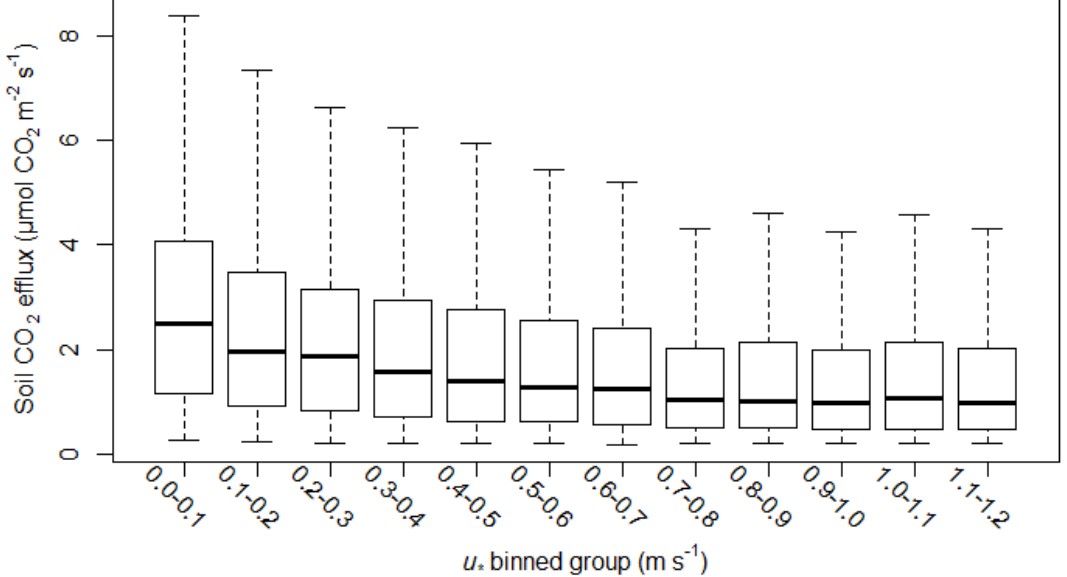

**Figure 4: Boxplot of mean hourly soil $CO_2$ effluxes for the one year campaign plotted against the binned groups of friction velocity ($u_*$).**





**Figure 5: Seasonally averaged diurnal patterns of soil CO$_2$ efflux (± standard deviation), at different friction velocity ($u_*$) threshold values, measured by the eight automated chambers for each of the 4 seasons. From the top, the four rows show the diurnal patterns for summer, autumn, winter and spring respectively. From the left, the five collars show the diurnal patterns for each season at no $u_*$ filtering, a $u_*$ threshold value of 0.1 m s$^{-1}$, a $u_*$ threshold value of 0.3 m s$^{-1}$, a $u_*$ threshold value of 0.5 m s$^{-1}$ and a $u_*$ threshold value of 0.7 m s$^{-1}$, respectively.**



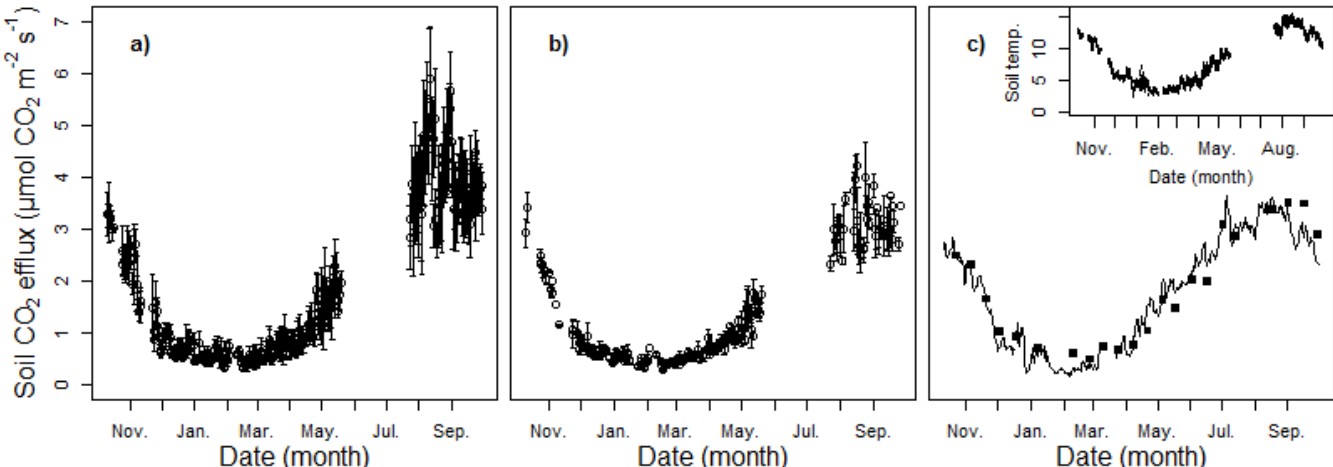

**Figure 6: Comparison of the courses of the mean daily soil $CO_2$ efflux throughout the year. (a) and (b) show mean daily soil $CO_2$ effluxes (± standard deviation) throughout the year measured by the eight automated soil chambers, without friction velocity ($u_*$) filtering and with a $u_*$ threshold values of 0.7 respectively. (c): The black dots show the mean soil $CO_2$ effluxes for each of the manual soil chamber campaigns, and the solid line shows the output of the empirical model based on these manual measurements. The course of soil temperature throughout the year at 5 cm depth measured at the eight automated chambers is shown on the inset in panel (c).**

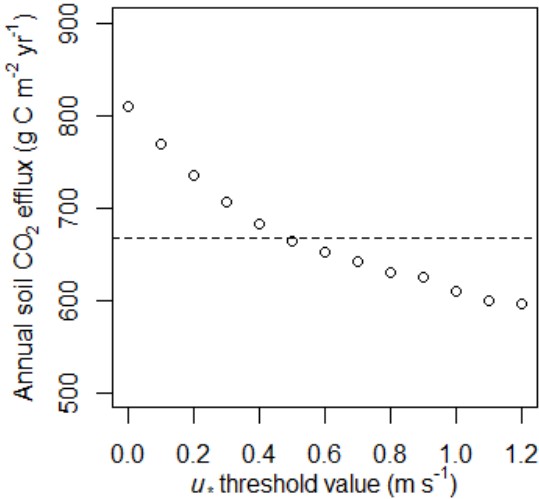

**Figure 7: Estimates of annual soil $CO_2$ efflux in response to increasing the friction velocity ($u_*$) threshold values for the automated chamber measurements during the one year campaign. The straight dashed line shows the annual soil $CO_2$ efflux of 666.6 g C m$^{-2}$ yr$^{-1}$ based on the manual measurements.**





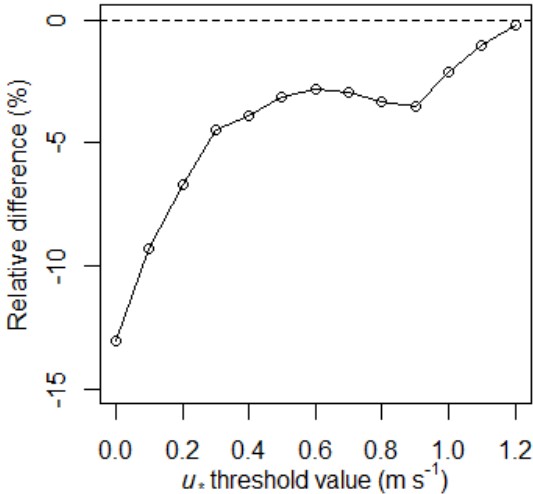

**Figure 8: Relative difference between the estimates of annual soil $CO_2$ efflux based on daytime (9–15 CET) data at different friction velocity ($u_*$) threshold values, compared to the estimates of annual soil $CO_2$ efflux based on data for the entire day at different friction velocity ($u_*$) threshold values.**

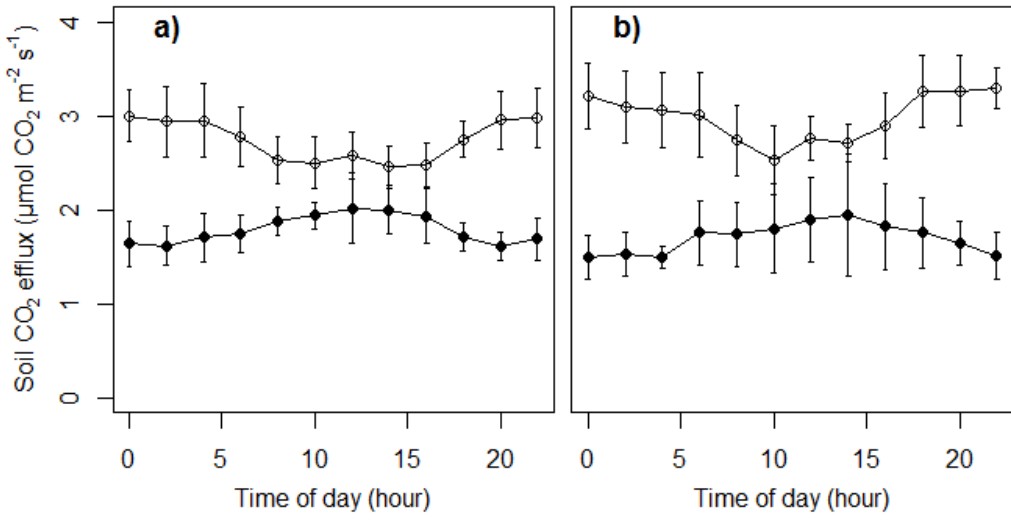

**Figure 9: Diurnal pattern of soil $CO_2$ efflux, measured by the automated chambers during the fan experiment, based on bi-hourly means (± standard deviation). (a) shows the diurnal pattern for half of the chambers with and without fans, where the first 10 days were with fans (filled circles) and the last 10 days were without fans (open circles). (b) shows the diurnal pattern with and without fans for the other half of the chambers, where the first 10 days were without fans (open circles) and the last 10 days were with fans (filled circles).**

