# Peer review of "Overestimation of closed chamber soil CO2 effluxes at low atmospheric turbulence"

_Biogeosciences, 2016_

## Referee Comment (RC1) · Anonymous Referee #1 · 1 Dec 2016

In this study, authors use an automated soil CO2 flux chamber to measure soil CO2 flux for over a year in a temperate Danish beech forest. The main finding of this study reported in this paper is that, with the closed chamber method for measuring soil CO2 flux, the soil CO2 flux could be significantly overestimated at night during lower atmospheric turbulent conditions. This is because under low turbulent condition, respired CO2 could accumulate near the soil surface, which slows down the CO2 diffusion out of soil. As a result, the CO2 concentration in the soil profile gets elevated. During the chamber-based measurement, the chamber movement breaks down this CO2 gradient. Thus, the CO2 diffusion gradient across the soil surface is much steeper than outside the chamber under natural condition, which lead to measured soil CO2 flux overestimated. This result presented in this paper and the explanation all make very good sense to me. The result is also consistent with an early publication (Schneider, et

al., 2009. Overestimation of CO2 respiration fluxes by the closed chamber method in low-turbulence nighttime conditions, J. Geophys. Res. Biogeosciences, 114(3), 1–10, doi:10.1029/2008JG000909).

The research site is ideal for this study, as the diurnal soil temperature variation is almost non-exist. Otherwise the high daytime soil temperature could easily override the impact of low turbulent condition on measured soil CO2 flux. In the paper, authors also try to use a fan for promoting the turbulent condition around the soil chamber and hoping the diffusion process for respired CO2 is always under steady-state condition. Their field dataset does show that this approach is very promising. One concern I have with this approach is that it might be possible for some fresh air being pushed into the soil profile by the fan, leading horizontal advection of air movement in the soil around the chamber, which would cause loss of respired CO2. This might be another reason why authors see lower measured soil CO2 flux (Fig. 9) even under high turbulent condition when the fan is used. Nevertheless, this paper could serve a starting point in the research community to stimulate more studies so a reliable and robust method to create a steady-state condition around a soil CO2 flux chamber will be available soon. Overall this manuscript is well written with a high quality of dataset. Conclusions are drawn based on defensible data.

---

## Referee Comment (RC2) · Anonymous Referee #2 · 12 Dec 2016

General Comments:

The manuscript from Brændholt et al. presents a very interesting study about the over-estimation of night-time soil respiration (Rs) measured by chambers, when low turbulence mixing occurs. The main finding of the study is that night-time Rs is inversely related to friction velocity, and filtering out data measured under a certain threshold of u* removes the differences between night-time and daytime Rs, underlying that the observed overestimation is due to poorly mixed air at the chamber level. This is a very important topic in the CO2 flux community also because chambers are often used to be compared with the underestimated nighttime CO2 flux by the eddy covariance method, but a gap still exists between the highly standardized procedures to process, check and filter EC data and the lack of common guidelines for checking the quality of chamber measurements. The manuscript is very well written and argued. The text is well structured and fluent, and the figures very clear. I think that there are some controversial points, e.g. the covariation of u* with some biological process in the diurnal cycle, such as translocation of assimilates and lag-effect in the root flux component (e.g. Heine-meyer al. 2011), the covariation of u* with temperature, the hysteresis between soil CO2 efflux and temperature. However, the authors well discussed many of them in the manuscript.

Specific points:

For personal experience, one important issue in chamber measurements is the soil collar insertion, not only regarding the depth and disturbances to different soil components of but also the collar height outside the soil surface. The 8100/8150 is designed to achieve a good mixing inside the chamber without using a fan. But without a fan inside the chambers, the collar should be set low to few cm (offet ∼5 cm), since too high collars could make the air mixing inside the chamber difficult during nighttime. The authors should specify in the manuscript not only the insertion depth but also the collar height and offset, and see if improvements could possibly be made by lowering the collar.

Related to the first point, a good solution for solve the night-time overestimation could be acting on the deadband in post processing. Indeed during nighttime the air mixing could be poor shortly after the chamber closes, but then a good mixing could be kept by the flow between the LI-8100 and chamber. Even if at lines 22-25 (pg 14) the authors explain that they checked different deadbands, they did not show results of this analysis. It could be interesting to see how a raw gradient appears during one daytime well-mixed measurement compared to one nighttime calm measurement and test if a larger deadband may in part improve the gradient fitting. Did the author try to recompute the whole dataset with a larger deadband (e.g. 50s) and see if the diurnal cycle is at least reduced? One limit could be the short measurement time used (90s). However, this could be a very important way, because if it is true that u* acts on the overestimation of chamber CO2 nighttime efflux, we would love to find a way not to

reject a big amount of data as for EC.

Depending on the air mixing conditions both linear and non-linear fit could perform better, for this reason the use of the best between the two methods for each single measure could be a solution to be tested.

The authors present an approach to keep an adequate mixing of air around the chambers by using table fans. Even if the use of fans represent an interesting and efficient test in this study, I'm wondering if instead the use of a "channelled" artificial wind in a very stratified atmosphere could instead drain out $CO_2$ from the near-chamber air volume, even if the authors mention this point, I think that the use or not of fan in chamber measurements should be better investigated before promoting its use.

Minor points

- I find the paragraphs at lines 23-31 and 32-34 pg. 3 too long: I suggest to focus only on the aims of the study and not on the many details concerning how the study is carried on that should be placed in the material and method section

-pg 5 – line 4: remove "which yielded a total of 52131..."

-pg 5 – line 22: "the current manuscript focuses on the potential error introduced by low turbulent..."

- pg 5 – line 25-27 this part could be better placed in section 2.2

- Fig. 1 probably a scatter plot of nighttime Rs versus u* values will better present the inverse relationship

---

## Author Comment (AC1) · 12 Dec 2016

We thank reviewer 1 for the review and comments, and we appreciate the reviewer's positive opinion towards the manuscript. We would like to comment on the following specific concern put forward by the reviewer:

"One concern I have with this approach is that it might be possible for some fresh air being pushed into the soil profile by the fan, leading horizontal advection of air movement in the soil around the chamber, which would cause loss of respired CO2. This might be another reason why authors see lower measured soil CO2 flux (Fig. 9) even under high turbulent condition when the fan is used."

We agree with the idea put forward by the reviewer, and think it falls well in line with the third explanation that we proposed for the lower fluxes with a fan, even under high

turbulence (see page 12, line 18 to page 13, line 4).

In the manuscript we discuss if using a fan not only eliminates the effect of low atmospheric turbulence, but also introduces a new potential bias of higher than ambient atmospheric turbulence, which could explain the lower fluxes seen even during day-time when atmospheric turbulence generally was high. We hypothesize that the lowering of chamber day-time fluxes might be due to lower wind speed in the chamber during a measurement compared to the outside, when the fan is used. The lower fluxes measured under these conditions may therefore be explained by the general relationship between wind speed and soil fluxes estimated with chambers reported in the literature.

---

## Referee Comment (RC3) · Anonymous Referee #3 · 14 Dec 2016

The paper presents the results of one year of automatic and manual chamber measurements of soil respiration conducted in a Danish beech forest and highlights the biases and uncertainties issues related to these measurements. Although I have some critical comments related to applied methodology and data processing I found this paper and dataset very valuable and important for chamber flux community and have to admit that it adds a new hints to a never-ending discussion about the quality of chamber fluxes, measurement protocols, data processing and data filtering. The paper is very well written and structured. However, I have some concerns and a number of suggestions, that I believe will improve this manuscript once addressed.

Major comments: 1. Page 3 line 32-33 – Maybe I understood wrongly, but from my point of view the hypotheses is stated wrongly or not precisely enough. There is written that overestimation of the CO2 fluxes during stable atmospheric conditions was due

to insufficient mixing of the air above the soil surface.. – do you mind the air inside the chamber headspace or in the open air? This should be clarified. If chamber headspace is considered I would avoid such hypotheses as it is discussed already in several papers that the effect of overestimation of nighttime fluxes is due to broken down the highly stratified layer of air inside the chamber headspace due to chamber movement at closure (Gorres et al. 2015) or air mixing in the chamber headspace (Schneider et al. 2009, Juszczak et al. 2012), which lead to change of predeployment steady state steep $CO_2$ concertation gradient above the soil due to air mixing in the chamber. However, if insufficient air mixing in the atmosphere is considered then I would avoid to promote any disruptions of this natural condition occurring during calm nights (by excessive artificial air mixing, as authors has suggested, as a possible solution to overcome the overestimation of night-time fluxes) as this again will artificially change the concentration gradients between the soil and the atmosphere and enhance emission of gases, which does not occur during nights with stable atmospheric conditions when the air is highly stratified and when the only process driving emission of $CO_2$ from the soil is diffusion.

2. Page 4 line 9 – could you please specify how dense the canopy is? This seems to be not so important for the paper but it helps to imagine how far a density of the forest canopy may impact the turbulences in the canopy, especially that the sonic anemometer was installed 43 meters above the soil surface. In the ecosystems with a short vegetation (e.g. grasslands), the u* filtering procedure can be applied to separate periods with calm and turbulent atmospheric conditions near the surface, but in a forest canopy with nearly 30 meters height of tree stands this might be more difficult, as under certain conditions the air might not be well mixed under canopy, although there are turbulences in the air above the canopy. Can you be sure that during turbulent conditions (at height of 43 m) there are still turbulences near the soil surface? Question is how dense the forest is? this will help to interpret the data you have. Maybe, it would help if you look for $CO_2$ storage and relate this amount to measured $FCO_2$ from EC? If storage is relatively high the air is not well mixed and may be stratified in the forest

canopy

I am not sure but considering the results you got from a fan experiment, which indicated that fluxes measured over the day and night were smaller when artificial mixing was applied (of course it introduce other changes in environment as is discussed in the paper), one may conclude that the near surface air in the forest floor is not well mixed even during the day when u* calculated based on measurements on 43 meters above this surface is above 1.2. Maybe this explain why there is no any diurnal cycle of CO2 fluxes detected by chamber measurements, especially that there is a weak diurnal pattern of soil temperature which drive respiration processes. This might be also the effect of the time lag between inputs of C via photosynthesis and Rs, as discussed in the paper, but maybe also factors indicated above may impact the measured fluxes for these conditions?

3. Page 5 line 25-30 regarding closure time (line5 of page 5) there is written that closure time was 90 and 150 sec in automated and manual chambers respectively, while for a fan experiment you extended the closure time to 5 minutes. Why the closure time was different? In case of manual measurements, the first 20 sec of data points were discarded (due to initial disturbances), while in case of a fan campaign you discarded first 60 seconds... this was due to time-leg the air came from the chamber to the analyzer? What was the length of tubes ? Was it the same for all chambers? Regarding fluxes please specify how the fluxes were calculated for automated chambers. Did you calculate fluxes with your R script or you relied on fluxes calculated by LICOR soft? Were the same quality criteria taken into account for all chambers (page 5 line 28-30)?. If not, is that mean that fluxes which does not pass the goodness-of-fit criteria were also taken for analyses? (in manual chambers number of fluxes is small hence the question is what kind of quality criteria were applied in this case). Another point is that the linear fitting was applied to calculate fluxes. This is absolutely correct if closure time is short, but in the case of a fan experiment your closure time was 2-3 times longer than in manual and automated chambers – as described above. I assume that if you

used the same small/short LICOR chambers there is for sure non-linear development of CO2 concertation in the chamber headspace. In this case we know that if linear fitting is applied to calculate fluxes then this will lead to significant flux underestimation. Did you consider this effect?

4. Please explain why different approaches were used to deliver annual CO2 effluxes for manual and automated chambers (Page 6 lines 7-23)? I do understand that the number of fluxes measured by automatic system is much higher than from manual one but still data coverage was 76% as you wrote in page 5 (272 days), besides there were for sure also gaps in daily data series. While, what you calculated for 12 sub-sets was just average daily flux. The missing data for period between 20 May and 22 June were estimated based on linear interpolation between hourly values, although you may use Lloyed and Taylor (1994) model to estimate missing fluxes much more accurate and with less uncertainty (from fig. 6 we know that there was clear seasonal pattern of soil temperature change). I am afraid that the approach used in the paper may bias estimates of annual fluxes. I am not sure if it is not too late now, but I would suggest to first model (with Lloyed and Taylor 1994 equation) the missing effluxes (for each automated chamber) to have a continuous data series of CO2 fluxes (looking for relationships between daily fluxes and T) and then by using u*, fluxes can be divided to 12 different sub-sets. This approach would be more accurate I assume, if you found any relationship between measured effluxes and soil temperature. Or, having so much data you may parameterize Lloyed and Taylor model for such datasets to calculate effluxes for the whole year based on the measured soil temperature. And then, Rreference (at 10oC) or annual fluxes for such subsets might be compared for different u* classes.

Considering above, please clarify how the annual fluxes were calculated. If the measured fluxes were divided to 12 different subsets (depending on u*), then for sure you had different numbers of fluxes for each class (please specify this information e.g. in Fig. 4). Please specify how the annual fluxes were calculated then? I understood that first data were filtered based on u* and 12 subsets were selected. That means that you

had for each day different number of fluxes for each chamber – these might be daytime and night-time fluxes or only daytime or only night time fluxes—how the daily fluxes were calculated then? Here, I would suggest to look for Rs vs T relationship for each subset and use e.g. Lloyed and Taylor function to model Rs for the whole year. If it is done in other way, I consider this incorrect, especially that we do not know how many fluxes were in each group and from which part of the day. This may impact results described in section 3.2 and 3.3., the whole discussion of results and might be critical point for the analyses presented in the paper

5. From data you presented is clear that CO2 effluxes are not following soil temperature changes over the day and they are inversely dependent on u*. Of course there is a weak diurnal change of soil temperature at 5 cm depth and mainly it appears during summer, but still I would expect higher fluxes in the afternoon when soil temperature reaches maximum. The filtering procedure you applied lead to significant reduction of the fluxes during nights and slight reduction of daytime fluxes (which also indicates that there were stable atmospheric conditions over the day, as also was indicated by a fan experiment), hence still Rs was the smallest in the afternoon (besides autumn Fig 5j). Considering above I am wondering whether the temperature sensors are installed correctly? Maybe they are too deep? Have you measured soil temperature at 2 cm depth? Is there any relationship between air temperature (near the surface) and soil temperature? If not, then I assume the trees canopy might be so dense that the soil surface is homogenously shadowed, but this may also mean that next to the surface (where chamber measurements are conducted) there might be not much turbulences (this should also be critically discussed in the paper). Please specify in methods how many temperature sensors were installed and how they were distributed over the site – were they installed in soil collars, next to, or few meters from? – this information is missing although it may help to understand why there is no correlation between measured fluxes and T (on daily basis). I assume this might be autotrophic respiration of tree roots which may dominate your Rs and may not be depend on temperature and if yes, the data analyses would be even more complicated.

6. The fan experiment described in the paper indicated that by mixing of the air during stable atmospheric conditions the near surface air is not so stratified and mixing of the air in the chamber does not lead to overestimation of the fluxes (but only if it is not too strong). I found this experiment interesting but from my point of view this should not be promoted as solution to overcome problems with nighttime chamber flux measurements. It is well known that during calm nights the only process driving emission from the soil is only diffusion, hence we should avoid to increase turbulence by excessive artificial mixing of the air nearby the chamber, as this change the emission of gases from the soil and will lead to increase fluxes which are much smaller when smaller gradients of $CO_2$ occur during calm nights. Another point is how to measure fluxes with chamber over calm conditions. Maybe the application of short chamber is a good solution (as proposed by Gorres et al 2016 – although with this short chambers other problems appear (as discussed in e.g. Pihlatie et al. 2013), but for sure one of the solution might be to reduce or eliminate air mixing in the chamber headspace to minimize disruption of stratified air in the chamber headspace. The artificial wind may cause also other problems, by changing air pressure inside the chamber by e.g. Venturi effect, or cause excessive latter fluxes which can impact measured chamber fluxes significantly.

Other minor comments and suggestions:

page 2 Line 18 – there was also a paper of Pumpanen et al. (2004) where rates of over- or underestimation of $CO_2$ fluxes measured by different chamebers are presented in the controlled conditions, it is worth to cite it here page 3 line 14-15, I would not agree that the mechanisms leading for flux overestimation are uncertain. They are well discussed in the cited papers and also in your paper (page 9-10) hence I would remove this sentence. Page 3 line 27 u∗ was measured continuously above the tree canopy – I am afraid that conditions under the tree canopy might be different than those above the canopy, especially that EC system is installed well above the forest canopy (43 m above the surface). See a comment above

Page 3, line 12 – please consider also whether to cite the paper of Juszczak et al.

(2012) in Polish Journal of Environmental Studies, who compared daytime and night-time Reco fluxes measured manually by chambers and filtered fluxes based on u*, and proved that when proper flux filtering (based on u*) is applied then there is no difference between day and night-time REco fluxes measured by chambers, while they are significantly overestimated when no filtering is applied. This is in agreement with your statements (page 11, lines 19-24).

Page 4 line 24-26 I am afraid that even if collars are close (for manual and automated chamber measurements) the fluxes are not comparable due to a high spatial hetero-geneity of soil respiration flux in the forest floor (due to many factors related to soil itself and distribution of roots, and hence different Ra and Ra/Rh ration)

Page 4 line 29 – if any plants appeared in the collar then they were cut or just removed with roots? What about surface layer then?

Page 5 line 15-20 please specify the height the fans are installed. The chamber you used are rather small/short and I assume fans were just above the soil surface? But this need to be written here

Page 5 line 10-20 Can you please clearly write in paragraph 2.3 what kind of chambers were used and why you extend the closure time in case of a fan experiment.

Page 5 line 31, write covariance instead of co-variance

Page 7 lines 25-30 are the rates of fluxes restricted to turbulent conditions, or average of all fluxes is considered? If yes, then it may explain differences you describe (auto-mated measurements combine data measured over the day and night, while manual measurements were conducted over the day (till 3 pm). If you compare fluxes which were filtered using u* then difference between manual and automated fluxes is not so big (Fig. 6b, c). In order to compare fluxes you should rather calculate average flux for fluxes measured in the same period from 9am to 15.

Page 12 lines 34 page 13 lines 1-5 – this was already suggested if I well remember in

Rochette and Hutchinson (2005),. They suggested that to avoid overmixing of the air in the chamber headspace the fan speed should be adjusted to outside wind speed.

---

## Short Comment (SC1) · 31 Dec 2016

I am very much impressed by the critical and careful analysis of CO2 efflux measurements with chambers. I think it is the first time that the problem has been clearly addressed on the basis of a good data set and some additional studies.

It is a pity that no additional sonic anemometer was installed in the trunk space, which was recently urgently recommended (Thomas et al., 2013). Therefore you are unable to indicate if the atmosphere above the canopy (where you measured the friction velocity) is coupled with the trunk space (Thomas and Foken, 2007). Your daily and annual cycle of the friction velocity is probably slightly modified by the significant daily and annual cycle of coupling (Foken et al., 2012; Jocher et al., 2017). Perhaps you should include in your recommendations a second sonic anemometer, which would control the

turbulent mixing (friction velocity, standard deviation of the vertical wind velocity) in the vicinity of the soil chambers. This may reduce the proposed very high u*-threshold.

The problem you addressed is not only related to the turbulent mixing or the friction velocity. Low friction velocities are often connected with stable stratification and the reason for very stable conditions near the surface is a cooling by a large longwave net radiation. In contrast to the natural condition, the longwave net radiation in a chamber is always nearly zero and therefore the stratification is always nearly neutral. This may also be a reason why a chamber under stable (night-time) conditions can overestimate the fluxes (Riederer et al., 2014). Our study was made above a meadow with much larger longwave net radiation than inside the canopy, but nevertheless the longwave radiation effect on chamber measurements should be discussed. Helpful would be four-component net radiometers above the forest and in the trunk space – perhaps a further recommendation for flux sites.

Finally, perhaps the following hypothesis could explain your findings: The chamber is like a "chimney", with nearly neutral stratification and high turbulent mixing. It is like a "convective hot spot" above the soil with stable stratification and nearly laminar flow in the surroundings. Because of the high carbon dioxide gas concentrations in the soil, a slight horizontal advection in the soil layer generates a high $CO_2$ flux in the chamber. This can also explain your found hysteresis, because this horizontal advective flow is slow. In the case of the fans in the surroundings of the chamber, you destroyed the chimney effect, because the well-mixed and neutral stratified area is much larger.

I think the paper should be accepted with the discussion of the two additional influencing factors in Sect. 4, but the authors should repeat the experiment at their well-equipped site with the additional instrumentation recommended above.

References

Foken, T., Meixner, F. X., Falge, E., Zetzsch, C., Serafimovich, A., Bargsten, A., Behrendt, T., Biermann, T., Breuninger, C., Dix, S., Gerken, T., Hunner, M., Lehmann-

[Figure]

Pape, L., Hens, K., Jocher, G., Kesselmeier, J., Lüers, J., Mayer, J. C., Moravek, A., Plake, D., Riederer, M., Rütz, F., Scheibe, M., Siebicke, L., Sörgel, M., Staudt, K., Trebs, I., Tsokankunku, A., Welling, M., Wolff, V., and Zhu, Z.: Coupling processes and exchange of energy and reactive and non-reactive trace gases at a forest site – results of the EGER experiment, Atmos. Chem. Phys., 12, 1923-1950, 10.5194/acp-12-1923-2012, 2012.

Jocher, G., Ottosson Löfvenius, M., De Simon, G., Hörnlund, T., Linder, S., Lundmark, T., Marshall, J., Nilsson, M. B., Näsholm, T., Tarvainen, L., Öquist, M., and Peichl, M.: Apparent winter CO2 uptake by a boreal forest due to decoupling, Agrical. Forest Meteorol., 232, 23-34, 10.1016/j.agrformet.2016.08.002, 2017.

Riederer, M., Serafimovich, A., and Foken, T.: Eddy covariance – chamber flux differences and its dependence on atmospheric conditions, Athmospheric Measurement Techniques, 7, 1057–1064, 10.5194/amt-7-1057-2014, 2014.

Thomas, C., and Foken, T.: Flux contribution of coherent structures and its implications for the exchange of energy and matter in a tall spruce canopy, Boundary-Layer Meteorol., 123, 317-337, 10.1007/s10546-006-9144-7, 2007.

Thomas, C. K., Martin, J. G., Law, B. E., and Davis, K.: Toward biologically meaningful net carbon exchange estimates for tall, dense canopies: Multi-level eddy covariance observations and canopy coupling regimes in a mature Douglas-fir forest in Oregon, Agrical. Forest Meteorol., 173, 14-27, 10.1016/j.agrformet.2013.01.001, 2013.

---

## Author Comment (AC2) · 30 Jan 2017

Dear Thomas Foken We thank you for showing an interest in the manuscript and for taking part in the discussion with helpful suggestions and comments. We provide the following response.

"It is a pity that no additional sonic anemometer was installed in the trunk space, which was recently urgently recommended (Thomas et al., 2013). Therefore you are unable to indicate if the atmosphere above the canopy (where you measured the friction velocity) is coupled with the trunk space (Thomas and Foken, 2007). Your daily and annual cycle of the friction velocity is probably slightly modified by the significant daily and annual cycle of coupling (Foken et al., 2012; Jocher et al., 2017). Perhaps you should include in your recommendations a second sonic anemometer, which would control the

turbulent mixing (friction velocity, standard deviation of the vertical wind velocity) in the vicinity of the soil chambers. This may reduce the proposed very high u*-threshold."

We agree that it could have been interesting to have a second anemometer closer to the ground to address the issues you mention. We will include a discussion of sonic anemometer height in section 4.5.

"The problem you addressed is not only related to the turbulent mixing or the friction velocity. Low friction velocities are often connected with stable stratification and the reason for very stable conditions near the surface is a cooling by a large longwave net radiation. In contrast to the natural condition, the longwave net radiation in a chamber is always nearly zero and therefore the stratification is always nearly neutral. This may also be a reason why a chamber under stable (night-time) conditions can overestimate the fluxes (Riederer et al., 2014). Our study was made above a meadow with much larger longwave net radiation than inside the canopy, but nevertheless the longwave radiation effect on chamber measurements should be discussed. Helpful would be four-component net radiometers above the forest and in the trunk space – perhaps a further recommendation for flux sites."

Thank you for pointing out another potential cause for stable conditions above the soil surface. We will include large longwave net radiation, as an explanation for stable stratification above the soil surface, alongside low turbulence in section 4.1.

"Finally, perhaps the following hypothesis could explain your findings: The chamber is like a "chimney", with nearly neutral stratification and high turbulent mixing. It is like a "convective hot spot" above the soil with stable stratification and nearly laminar flow in the surroundings. Because of the high carbon dioxide gas concentrations in the soil, a slight horizontal advection in the soil layer generates a high $CO_2$ flux in the chamber. This can also explain your found hysteresis, because this horizontal advective flow is slow. In the case of the fans in the surroundings of the chamber, you destroyed the chimney effect, because the well-mixed and neutral stratified area is much larger."

We agree that a chamber is like a "convective hot spot", as you phrase it, during stable stratification in the surroundings, and that using the fans destroy the chimney effect.

"I think the paper should be accepted with the discussion of the two additional influencing factors in Sect. 4, but the authors should repeat the experiment at their welle-quipped site with the additional instrumentation recommended above."

Thank you. It could indeed be very interesting to repeat the experiment with the additional instrumentation you recommend, and we hope we will get the chance to do so in the future.

---

## Author Comment (AC3) · 30 Jan 2017

We thank reviewer 2 for taking the time to review the manuscript and for providing helpful suggestions. We provide the following response to the reviewer's questions, suggestions and comments.

"Specific points: For personal experience, one important issue in chamber measurements is the soil collar insertion, not only regarding the depth and disturbances to different soil components of but also the collar height outside the soil surface. The 8100/8150 is designed to achieve a good mixing inside the chamber without using a fan. But without a fan inside the chambers, the collar should be set low to few cm (offet âĹij5 cm), since too high collars could make the air mixing inside the chamber difficult during nighttime. The authors should specify in the manuscript not only the insertion

depth but also the collar height and offset, and see if improvements could possibly be made by lowering the collar."

We will include information of soil collar height in section 2.2. The average collar height for the 8 soil chambers was 5.2 cm with the individual collar heights ranging from 4.5 to 5.8 cm. This seems to fall well in line with low collar height required from the reviewer's experience.

"Related to the first point, a good solution for solve the night-time overestimation could be acting on the deadband in post processing. Indeed during nighttime the air mixing could be poor shortly after the chamber closes, but then a good mixing could be kept by the flow between the LI-8100 and chamber. Even if at lines 22-25 (pg 14) the authors explain that they checked different deadbands, they did not show results of this analysis. It could be interesting to see how a raw gradient appears during one daytime well-mixed measurement compared to one nighttime calm measurement and test if a larger deadband may in part improve the gradient fitting. Did the author try to recompute the whole dataset with a larger deadband (e.g. 50s) and see if the diurnal cycle is at least reduced? One limit could be the short measurement time used (90s). However, this could be a very important way, because if it is true that u* acts on the overestimation of chamber CO2 nighttime efflux, we would love to find a way not to reject a big amount of data as for EC. Depending on the air mixing conditions both linear and non-linear fit could perform better, for this reason the use of the best between the two methods for each single measure could be a solution to be tested."

Based on results by Lai et al. (2012), we decided to look at different dead bands up to 40 s, to see if dead band had an influence on the overestimation. The overestimation of fluxes at low u*, however, persisted independent of dead band. We therefore decided just to mention these results in the paper, instead of showing them, which would add considerable length to the manuscript. We also inspected the increase in CO2 concentration during chamber closure both during high and low turbulence. However, the increase seemed to follow a pattern expected from flux theory during both high and low

turbulence. We also tried to see if removing fluxes at higher r2 values of the fit to the increase in CO2 concentration, during chamber closure, could remove overestimated fluxes at low atmospheric turbulence (see line 30, page 13 to line 4, page 14). However, a similar response of u* was still seen, indicating that the increase in chamber CO2 concentration followed an expected pattern that let to high r2 values, even though the measurement resulted in an overestimated flux. The 90 s chamber closure time was indeed a limit to investigating flux overestimation at low turbulence at much longer dead bands, and a longer chamber closure time could beneficially be used for future studies.

"The authors present an approach to keep an adequate mixing of air around the chambers by using table fans. Even if the use of fans represent an interesting and efficient test in this study, I'm wondering if instead the use of a "channelled" artificial wind in a very stratified atmosphere could instead drain out CO2 from the near-chamber air volume, even if the authors mention this point, I think that the use or not of fan in chamber measurements should be better investigated before promoting its use."

We agree that it is possible that using fans can drain out CO2 from the near-chamber air volume. We also agree that using fans for chamber measurements need to be better investigated before this approach can be promoted. This is in line with our statement on line 15, page 15 where we write that "Additional studies are needed to further explore this approach".

"Minor points - I find the paragraphs at lines 23-31 and 32-34 pg. 3 too long: I suggest to focus only on the aims of the study and not on the many details concerning how the study is carried on that should be placed in the material and method section"

We agree. We will shorten the paragraphs to focus only on the aims of the study.

"-pg 5 – line 4: remove "which yielded a total of 52131..."

We will remove it.

[Figure]

"-pg 5 – line 22: "the current manuscript focuses on the potential error introduced by low turbulent..."

We will change "in" to "on".

"- pg 5 – line 25-27 this part could be better placed in section 2.2"

We agree to move the lines. We will suggest moving the first sentence, concerning the one year campaign, to section 2.2., and the second sentence, concerning the fan experiment, to section 2.3.

"- Fig. 1 probably a scatter plot of nighttime Rs versus u* values will better present the inverse relationship"

Thank you for the suggestion. We initially tried a scatter plot. However, the large amount of data resulted in a plot overloaded with data points. Instead, we provide a boxplot of mean hourly soil $CO_2$ effluxes plotted against binned groups of u* in figure 4, which we think better presented the inverse relationship.

References: Lai, D. Y. F., Roulet, N. T., Humphreys, E. R., Moore, T. R., and Dalva, M.: The effect of atmospheric turbulence and chamber deployment period on autochamber $CO_2$ and $CH_4$ flux measurements in an ombrotrophic peatland, Biogeosciences, 9(8), 3305–3322, doi:10.5194/bg-9-3305-2012, 2012.

---

## Author Comment (AC4) · 30 Jan 2017

We thank reviewer 3 for the taking the time to review the manuscript and for providing helpful comments and suggestions. We provide the following response to the reviewer.

"Major comments: 1. Page 3 line 32-33 – Maybe I understood wrongly, but from my point of view the hypotheses is stated wrongly or not precisely enough. There is written that overestimation of the CO2 fluxes during stable atmospheric conditions was due to insufficient mixing of the air above the soil surface. – do you mind the air inside the chamber headspace or in the open air? This should be clarified. If chamber headspace is considered I would avoid such hypotheses as it is discussed already in several papers that the effect of overestimation of nighttime fluxes is due to broken down the highly stratified layer of air inside the chamber headspace due to chamber movement

at closure (Gorres et al. 2015) or air mixing in the chamber headspace (Schneider et al. 2009, Juszczak et al. 2012), which lead to change of predeployment steady state steep CO2 concertation gradient above the soil due to air mixing in the chamber. However, if insufficient air mixing in the atmosphere is considered then I would avoid to promote any disruptions of this natural condition occurring during calm nights (by excessive artificial air mixing, as authors has suggested, as a possible solution to overcome the overestimation of night-time fluxes) as this again will artificially change the concentration gradients between the soil and the atmosphere and enhance emission of gases, which does not occur during nights with stable atmospheric conditions when the air is highly stratified and when the only process driving emission of CO2 from the soil is diffusion."

We agree that the hypothesis on Page 3 line 32-33 could have been stated clearer. We will correct this, so that it is clear that we mean that overestimation of the CO2 fluxes during stable atmospheric conditions is due to insufficient mixing of open air above the soil surface. We don't fully follow the distinction made by the reviewer between insufficient mixing of air in the chamber headspace and in the free air above the soil surface. Both the effect seen by (Görres et al. 2016) and (Schneider et al. 2009) as well as in our study is due to insufficient mixing of air prior to the measurements and not due to insufficient mixing in the chamber headspace. We discuss how to get unbiased closed chamber measurements during low turbulence (section 4.5, page 13). We argue that mixing of chamber air is a requirement for closed chamber measurements to work, i.e. we want to measure diffusion from the soil to the chamber atmosphere and not between layers within the chamber atmosphere. This makes it difficult to get a reliable flux estimate, as shown in our study, whenever the free air above the soil surface is not well mixed. We agree with the reviewer that using a fan, as well as performing a chamber measurement during low turbulence, will change the concentration gradient and enhance the emission. We think it is important to state that this is without a chamber on the soil. The undisturbed soil CO2 efflux is thus lower during calm nights. Soil respiration may, however, be identical to a period with well mixed condition. By

providing mixing of free air with a fan, there is a closer link between soil respiration and soil CO2 efflux, and maybe also between the apparent soil CO2 efflux measured by the chamber and soil respiration.

"2. Page 4 line 9 – could you please specify how dense the canopy is? This seems to be not so important for the paper but it helps to imagine how far a density of the forest canopy may impact the turbulences in the canopy, especially that the sonic anemometer was installed 43 meters above the soil surface. In the ecosystems with a short vegetation (e.g. grasslands), the u* filtering procedure can be applied to separate periods with calm and turbulent atmospheric conditions near the surface, but in a forest canopy with nearly 30 meters height of tree stands this might be more difficult, as under certain conditions the air might not be well mixed under canopy, although there are turbulences in the air above the canopy. Can you be sure that during turbulent conditions (at height of 43 m) there are still turbulences near the soil surface? Question is how dense the forest is? this will help to interpret the data you have. Maybe, it would help if you look for CO2 storage and relate this amount to measured FCO2 from EC? If storage is relatively high the air is not well mixed and may be stratified in the forest canopy"

The annual duration of canopy cover is 180 days with a peak LAI of 5.0. The number of tree stems per hectare is 266. We will include this information on page 4. We cannot be sure that there at all times also are turbulence near the soil surface whenever there are turbulent conditions above the canopy. Earlier unpublished studies show that there is some degree of correlation between turbulence above the canopy and at the soil surface. We appreciate the suggestion regarding CO2 storage. That might have provided some information

"I am not sure but considering the results you got from a fan experiment, which indicated that fluxes measured over the day and night were smaller when artificial mixing was applied (of course it introduce other changes in environment as is discussed in the paper), one may conclude that the near surface air in the forest floor is not well

mixed even during the day when u* calculated based on measurements on 43 meters above this surface is above 1.2. Maybe this explain why there is no any diurnal cycle of CO2 fluxes detected by chamber measurements, especially that there is a weak diurnal pattern of soil temperature which drive respiration processes. This might be also the effect of the time lag between inputs of C via photosynthesis and Rs, as discussed in the paper, but maybe also factors indicated above may impact the measured fluxes for these conditions?"

We agree that it is possible that the near surface air in the forest floor is not well mixed even during day-time. We did detect a slight diurnal cycle in soil CO2 effluxes when a fan was applied, with peak CO2 effluxes during day-time (Fig. 9 page 24), which was in contrast to the diurnal cycle seen during summer of the one year campaign where the highest fluxes were seen during night-time (Fig. 5a page 22).

"3. Page 5 line 25-30 regarding closure time (line5 of page 5) there is written that closure time was 90 and 150 sec in automated and manual chambers respectively, while for a fan experiment you extended the closure time to 5 minutes. Why the closure time was different? In case of manual measurements, the first 20 sec of data points were discarded (due to initial disturbances), while in case of a fan campaign you discarded first 60 seconds: : : this was due to time-leg the air came from the chamber to the analyzer? What was the length of tubes ? Was it the same for all chambers? Regarding fluxes please specify how the fluxes were calculated for automated chambers. Did you calculate fluxes with your R script or you relied on fluxes calculated by LICOR soft? Were the same quality criteria taken into account for all chambers (page 5 line 28-30)?. If not, is that mean that fluxes which does not pass the goodness-of-fit criteria were also taken for analyses? (in manual chambers number of fluxes is small hence the question is what kind of quality criteria were applied in this case). Another point is that the linear fitting was applied to calculate fluxes. This is absolutely correct if closure time is short, but in the case of a fan experiment your closure time was 2-3 times longer than in manual and automated chambers – as described above. I assume that if you

used the same small/short LICOR chambers there is for sure non-linear development of CO2 concertation in the chamber headspace. In this case we know that if linear fitting is applied to calculate fluxes then this will lead to significant flux underestimation. Did you consider this effect?"

We used a short closure time of 90 s for the automated chambers measurements during the one year campaign. We wanted to keep the closure time short to get a higher number of flux measurements. The 90 s was found to be sufficient to ensure a high enough increase in CO2 concentration during chamber closure time to provide a solid flux calculation. For the manual measurements with the 10 cm survey chamber we used a longer closure time of 150 s. Experiences from previous years of measurements have shown that it can be difficult to achieve a high flux coefficient of variance (as provided by the LI-COR software immediately following a measurement) during winter when fluxes are low. We have therefore found that a longer chamber closure time is required. We used a longer enclosure time and dead band for the automated measurements during the fan experiment because an external gas analyzer was connected to the LI-8100/LI-8150 system in relation to another experiment. The external gas analyzer required a bigger difference in CO2 concentration during a chamber measurement for precise measurements. This is why a longer chamber closure was used. We found that due to the extra volume of the external gas analyzer, a longer period for the air to be mixed and to stabilize at the beginning of a chamber measurement was required. A longer dead band was therefore used. The tubes between the LI-8150 multiplexer and the chambers were 10-15 m long. The different tube lengths were accounted for in the flux calculation. We calculated fluxes using R. The linear fluxes were calculated using the lm function and the non-linear fluxes were calculated by fitting the non-linear equation suggested by Hutchinson and Mosier (1981) using the nlsLM function. The same goodness-of-fit criteria were used for all flux measurements from the automated chambers. I.e. fluxes with an r2 < 0.95 of the linear regression were removed from further analysis (page 5, line 28). For the manual measurements the quality control was done in the field directly following a measurement. If the flux CV

was higher than 1.4, the measurement was discarded and an extra measurement was performed on the soil collar. We will specify this in the manuscript. We are fully aware of, and agree with you on the general risk of flux underestimation when applying linear regressions to observations (e.g. see Line 20-25, page 2). However, non-linear curve fitting also introduces new potential errors and biases and potentially more variable output. As we focus the current paper on the potential overestimation of fluxes during low turbulence we decided to use linear regression to get robust flux estimates.

"4. Please explain why different approaches were used to deliver annual CO2 effluxes for manual and automated chambers (Page 6 lines 7-23)? I do understand that the number of fluxes measured by automatic system is much higher than from manual one but still data coverage was 76% as you wrote in page 5 (272 days), besides there were for sure also gaps in daily data series. While, what you calculated for 12 subsets was just average daily flux. The missing data for period between 20 May and 22 June were estimated based on linear interpolation between hourly values, although you may use Lloyed and Taylor (1994) model to estimate missing fluxes much more accurate and with less uncertainty (from fig. 6 we know that there was clear seasonal pattern of soil temperature change). I am afraid that the approach used in the paper may bias estimates of annual fluxes. I am not sure if it is not too late now, but I would suggest to first model (with Lloyed and Taylor 1994 equation) the missing effluxes (for each automated chamber) to have a continuous data series of CO2 fluxes (looking for relationships between daily fluxes and T) and then by using u*, fluxes can be divided to 12 different sub-sets. This approach would be more accurate I assume, if you found any relationship between measured effluxes and soil temperature. Or, having so much data you may parameterize Lloyed and Taylor model for such datasets to calculate effluxes for the whole year based on the measured soil temperature. And then, Rreference (at 10oC) or annual fluxes for such subsets might be compared for different u* classes."

We used the empirical model by Lloyd and Taylor (1994) that was parameterized by the manual measurements to deliver the annual soil CO2 effluxes as this is a common

way of getting a continuous set of data based on soil CO2 effluxes measured at a low temporal scale. However, the high data coverage of automated measurements allowed us to calculate an annual soil CO2 effluxes based on these measurements directly. We could thus compare two very different approaches. We don't see how calculating an annual flux directly from the automated measurement and filling the data gap with linear interpolation can lead to any directional bias in the annual flux estimate. However we agree that it is possible that a parameterized Lloyd and Taylor could have been used to estimated missing flux values better than linear interpolation. However, we don't see that it could have influenced the difference in the annual calculated fluxes estimates between the different data sets with different u* filters.

"Considering above, please clarify how the annual fluxes were calculated. If the measured fluxes were divided to 12 different subsets (depending on u*), then for sure you had different numbers of fluxes for each class (please specify this information e.g. in Fig. 4). Please specify how the annual fluxes were calculated then? I understood that first data were filtered based on u* and 12 subsets were selected. That means that you had for each day different number of fluxes for each chamber – these might be daytime and night-time fluxes or only daytime or only night time fluxesâËŸAËĞThow the daily fluxes were calculated then? Here, I would suggest to look for Rs vs T relationship for each subset and use e.g. Lloyed and Taylor function to model Rs for the whole year. If it is done in other way, I consider this incorrect, especially that we do not know how many fluxes were in each group and from which part of the day. This may impact results described in section 3.2 and 3.3., the whole discussion of results and might be critical point for the analyses presented in the paper"

The different subsets with different u* filters did indeed have a different number of fluxes. Increasing the u* filter led to a lower number of fluxes. We will specify the number of fluxes in each of the binned groups of u* in figure 4. It is true that u* filtering removed fluxes, such that a single day might contain day-time and night-time fluxes or only day-time or night-time fluxes as you describe. However, we accounted for

this when calculating the annual fluxes, such that a calculated annual flux were not potentially biased by days containing e.g. only night-time values. The annual soil $CO_2$ efflux was for each of the 12 subsets calculated as described in line 7-9, page 6. First, a mean flux was calculated for each of the 24 hours of the day. This was done on a monthly basis. This ensured equal weight to all times of the day, even when the u* filtering had removed e.g. night-time fluxes only for specific days of that month. From the diurnal pattern, the mean daily flux on a monthly basis was calculated as the average of the 24 flux values across the day. From the mean daily flux on a monthly basis, the flux for the entire month was calculated by summation. From this the sum of the monthly fluxes gave the annual flux. We consider the method used to give a good calculation of annual soil $CO_2$ efflux for each of the 12 u* subsets. Thus we don't see that any other method (ours including) than an Rs vs T relationship function would be incorrect as stated by the reviewer.

"5. From data you presented is clear that $CO_2$ effluxes are not following soil temperature changes over the day and they are inversely dependent on u*. Of course there is a weak diurnal change of soil temperature at 5 cm depth and mainly it appears during summer, but still I would expect higher fluxes in the afternoon when soil temperature reaches maximum. The filtering procedure you applied lead to significant reduction of the fluxes during nights and slight reduction of daytime fluxes (which also indicates that there were stable atmospheric conditions over the day, as also was indicated by a fan experiment), hence still Rs was the smallest in the afternoon (besides autumn Fig 5j). Considering above I am wondering whether the temperature sensors are installed correctly? Maybe they are too deep? Have you measured soil temperature at 2 cm depth? Is there any relationship between air temperature (near the surface) and soil temperature? If not, then I assume the trees canopy might be so dense that the soil surface is homogenously shadowed, but this may also mean that next to the surface (where chamber measurements are conducted) there might be not much turbulences (this should also be critically discussed in the paper). Please specify in methods how many temperature sensors were installed and how they were distributed over the site

– were they installed in soil collars, next to, or few meters from? – this information is missing although it may help to understand why there is no correlation between measured fluxes and T (on daily basis). I assume this might be autotrophic respiration of tree roots which may dominate your Rs and may not be depend on temperature and if yes, the data analyses would be even more complicated."

We will specify the number of temperature sensors and how they were distributed in the methods section. Soil temperature in figure 3 is the average of 6 soil probes. They are distributed close to the soil chambers such that no soil chamber is further away than 10 m from an individual soil probe. We inserted each sensor 5 cm into the soil and we believe that they are inserted correctly, and not deeper than 5 cm. We did not measure soil temperature at 2 cm depth, but we agree that this could have been interesting to see if this would closer match soil CO2 effluxes. Air temperature at the site is generally highest in the early afternoon and lowest in the morning just around sunrise. The peak in soil temperature peaks a few hours later, which is to be expected from the heat capacity of the soil. We agree that lag times for autotrophic respiration can complicate matters and make the soil respiration look independent from soil temperature.

"6. The fan experiment described in the paper indicated that by mixing of the air during stable atmospheric conditions the near surface air is not so stratified and mixing of the air in the chamber does not lead to overestimation of the fluxes (but only if it is not too strong). I found this experiment interesting but from my point of view this should not be promoted as solution to overcome problems with nighttime chamber flux measurements. It is well known that during calm nights the only process driving emission from the soil is only diffusion, hence we should avoid to increase turbulence by excessive artificial mixing of the air nearby the chamber, as this change the emission of gases from the soil and will lead to increase fluxes which are much smaller when smaller gradients of CO2 occur during calm nights. Another point is how to measure fluxes with chamber over calm conditions. Maybe the application of short chamber is a good solution (as proposed by Gorres et al 2016 – although with this short chambers other problems

appear (as discussed in e.g. Pihlatie et al. 2013), but for sure one of the solution might be to reduce or eliminate air mixing in the chamber headspace to minimize disruption of stratified air in the chamber headspace. The artificial wind may cause also other problems, by changing air pressure inside the chamber by e.g. Venturi effect, or cause excessive latter fluxes which can impact measured chamber fluxes significantly."

We don't promote the use of fans. But we think the method has potential, why we in line 15, page 15 write that "Additional studies are needed to further explore this approach". How to get reliable flux measurements during low turbulence is indeed an interesting question that we also discuss in section 4.5, page 15. Regarding the Venturi effect, the pressure vent on the LICOR chambers is designed exactly to overcome and quickly release any pressure changes caused by changes in wind outside the closed chamber (Xu et al. 2006).

"Other minor comments and suggestions: page 2 Line 18 – there was also a paper of Pumpanen et al. (2004) where rates of over- or underestimation of $CO_2$ fluxes measured by different chamebers are presented in the controlled conditions, it is worth to cite it here page 3 line 14-15, I would not agree that the mechanisms leading for flux overestimation are uncertain. They are well discussed in the cited papers and also in your paper (page 9-10) hence I would remove this sentence. Page 3 line 27 uâ′L°U was measured continuously above the tree canopy – I am afraid that conditions under the tree canopy might be different than those above the canopy, especially that EC system is installed well above the forest canopy (43 m above the surface). See a comment above"

We will cite Pumpanen et al. (2004) on page page 2 Line 18. We will delete the sentence on page 3, line 14, concerning the uncertainty of the mechanisms leading to flux overestimation.

"Page 3, line 12 – please consider also whether to cite the paper of Juszczak et al. (2012) in Polish Journal of Environmental Studies, who compared daytime and nighttime Reco fluxes measured manually by chambers and filtered fluxes based on u*, and proved that when proper flux filtering (based on u*) is applied then there is no difference between day and night-time REco fluxes measured by chambers, while they are significantly overestimated when no filtering is applied. This is in agreement with your statements (page 11, lines 19-24)."

Thank you for suggesting the Juszczak et al. (2012) paper. We see that it contains relevant information and we will cite it on page 3, line 12. We will also refer to the paper following our discussion on page 11, lines 19-23.

"Page 4 line 24-26 I am afraid that even if collars are close (for manual and automated chamber measurements) the fluxes are not comparable due to a high spatial heterogeneity of soil respiration flux in the forest floor (due to many factors related to soil itself and distribution of roots, and hence different Ra and Ra/Rh ration)"

We agree that even soil collars very close to each other are not completely comparable due to the high spatial heterogeneity of soil respiration.

"Page 4 line 29 – if any plants appeared in the collar then they were cut or just removed with roots? What about surface layer then?"

The litter on the surface was kept intact, including new litter in the autumn. Tree branches that fell on the soil collars were removed. Otherwise these could physically prevent the soil chambers from closing. New plant shoots were removed by hand. The soil collars were checked at least every two weeks and new plant shoots never reached a height of more than a few cm. The roots (or the biggest part of the root) were most often removed as well when the plant was pulled out.

"Page 5 line 15-20 please specify the height the fans are installed. The chamber you used are rather small/short and I assume fans were just above the soil surface? But this need to be written here"

The fans were installed such that the middle of the fan was 30 cm above the soil

surface. We will include this information on page 5.

"Page 5 line 10-20 Can you please clearly write in paragraph 2.3 what kind of chambers were used and why you extend the closure time in case of a fan experiment."

We will specify the specific chamber models used and why the extended closure time was used. See also our response to "Major comment 3".

"Page 5 line 31, write covariance instead of co-variance"

We will change "co-variance" to "covariance"

"Page 7 lines 25-30 are the rates of fluxes restricted to turbulent conditions, or average of all fluxes is considered? If yes, then it may explain differences you describe (automated measurements combine data measured over the day and night, while manual measurements were conducted over the day (till 3 pm). If you compare fluxes which were filtered using u* then difference between manual and automated fluxes is not so big (Fig. 6b, c). In order to compare fluxes you should rather calculate average flux for fluxes measured in the same period from 9am to 15."

The automated fluxes on page 7, line 25-30 are without a u* filter applied. We compare fluxes measured for the entire day (section 3.2, page 8) and at day-time only (section 3.3, page 8).

"Page 12 lines 34 page 13 lines 1-5 – this was already suggested if I well remember in Rochette and Hutchinson (2005),. They suggested that to avoid overmixing of the air in the chamber headspace the fan speed should be adjusted to outside wind speed."

Thank you for the suggestion. We checked Rochette and Hutchinson (2005) and they indeed suggest matching chamber head space mixing intensities with pre-deployment conditions to avoid biased flux estimates. We will include a reference to Rochette and Hutchinson (2005) on page 12.

References:

Görres, C.-M., Kammann, C., and Ceulemans, R.: Automation of soil flux chamber measurements: potentials and pitfalls, Biogeosciences, 13, 1949-1966, doi: 10.5194/bg-13-1949-2016, 2016.

Hutchinson, G. L. and Mosier, A. R.: Improved soil cover method for field measurements of nitrous oxide fluxes, Soil Sci. Soc. Am. J., 45, 311–316, 1981.

Juszczak, R., Acosta, M. and Olejnik, J.: Comparison of daytime and nighttime ecosystem respiration measured by the closed chamber technique on a temperate mire in Poland, Polish J. Environ. Stud., 21(3), 643–658, 2012. Pumpanen, J., Westman, C. J. and Ilvesniemi, H.: Soil CO2 efflux from a podzolic forest soil before and after forest clear-cutting and site preparation, Boreal Environ. Res., 9(3), 199–212, 2004.

Rochette, P. and Hutchinson, G. L.: Measurement of Soil Respiration in situ: Chamber Techniques. In: Hatfield J.L., Baker J.M. (Eds.), Micrometeorology in Agricultural Systems, ASA-CSSA-SSSA, Agron. Monogr., 47, pp 247– 286, 2005.

Xu, L., Furtaw, M. D., Madsen, R. A., Garcia, R. L., Anderson, D. J., and McDermitt, D. K.: On maintaining pressure equilibrium between a soil CO2 flux chamber and the ambient air, J. Geophys. Res., 111, D08S10, doi:10.1029/2005JD006435, 2006.

---

## Author Response (AR1)

Dear Editor

Thank you for the opportunity to provide a revised version of our manuscript. We have in the following specified the changes made in accordance with the suggestions from each reviewer. We have not included the discussion again from our first response to the reviewers that did not require changes to the manuscript.

All changes to the manuscript can be seen in the marked-up version of the manuscript. The page and line numbers in our reply to the reviewers refer to the marked-up manuscript.

Furthermore, as suggested we have included supplementary information containing the results of the analysis of calculating soil $CO_2$ effluxes using different dead bands and calculation methods. However, a similar pattern of overestimation of soil $CO_2$ effluxes during low $u_*$, as reported in the manuscript for linearly calculated effluxes with a dead-band of 20 s, was seen. Thus neither dead band nor flux calculation method could eliminate the effect of low $u_*$ on soil $CO_2$ effluxes in our study.

We hope that the revised manuscript will be suitable for publication in Biogeosciences.

Andreas Brændholt,
on behalf of the authors

**Reply to reviewer 1**

We have not made any changes to the manuscript following the reviewer's comments.

**Reply to reviewer 2**

We have made the following changes to the manuscript in accordance with the reviewer's suggestions:

*"Specific points:*
*For personal experience, one important issue in chamber measurements is the soil collar insertion, not only regarding the depth and disturbances to different soil components of but also the collar height outside the soil surface. The 8100/8150 is designed to achieve a good mixing inside the chamber without using a fan. But without a fan inside the chambers, the collar should be set low to few cm (offet ~5 cm), since too high collars could make the air mixing inside the chamber difficult during nighttime. The authors should specify in the manuscript not only the insertion depth but also the collar height and offset, and see if improvements could possibly be made by lowering the collar."*

We have added the following to section 2.2., page 4, line 28-29:
"The average soil collar height was 5.2 cm with the individual collar heights ranging from 4.5 to 5.8 cm."

*"Related to the first point, a good solution for solve the night-time overestimation could be acting on the deadband in post processing. Indeed during nighttime the air mixing could be poor shortly after the chamber closes, but then a good mixing could be kept by the flow between the LI-8100 and chamber. Even if at lines 22-25 (pg 14) the authors explain that they checked different deadbands, they did not show results of this analysis. It could be interesting to see how a raw gradient appears during one daytime well-mixed measurement compared to one nighttime calm measurement and test if a larger deadband may in part improve the gradient fitting. Did the author try to recompute the whole dataset with a larger deadband (e.g. 50s) and see if the diurnal cycle is at least reduced? One limit could be the short measurement time used (90s). However, this could*

*be a very important way, because if it is true that u\* acts on the overestimation of chamber CO2 nighttime efflux, we would love to find a way not to reject a big amount of data as for EC.*
*Depending on the air mixing conditions both linear and non-linear fit could perform better, for this reason the use of the best between the two methods for each single measure could be a solution to be tested."*

We have included the results of the analysis of dead bands of 10, 20, 30 and 40 seconds for linearly calculated effluxes and non-linearly calculated effluxes, respectively, in the supplementary information.
We have deleted "data not shown" and replaced it with "see supplementary information" on page 15, line 26-27, following the discussion concerning different dead bands.

*"Minor points*
*- I find the paragraphs at lines 23-31 and 32-34 pg. 3 too long: I suggest to focus only on the aims of the study and not on the many details concerning how the study is carried on that should be placed in the material and method section"*

We have shortened the paragraphs to focus only on the aims of the study. The paragraphs on page 3, line 25 to page 4, line 3 now reads:
"Our study had two aims. The first aim was to quantify the effect of $u_*$ on automated closed chamber soil $CO_2$ effluxes in the short-term (i.e. effect on diurnal fluxes) and in the long-term (i.e. effect on annual estimates of $CO_2$ efflux). The second aim was to test the hypothesis that the overestimation of soil $CO_2$ effluxes during low $u_*$ was due to insufficient mixing of the open air above the soil surface and to test if unbiased soil $CO_2$ efflux measurements could be achieved during low $u_*$ by artificially inducing mixing of the air around the soil chambers by a fan. "

*"-pg 5 – line 4: remove "which yielded a total of 52131..."*

We have removed ", which yielded a total of 52131 individual unique chamber efflux measurements" on page 5, line 8-9.

*"-pg 5 – line 22: "the current manuscript focuses on the potential error introduced by low turbulent..."*

We have changed "in" to "on" on page 6, line 2.

*"- pg 5 – line 25-27 this part could be better placed in section 2.2"*

We have moved and rearranged the first sentence, concerning the one year campaign, to section 2.2, page 5, line 9-10. It now reads:
"Chamber closure time was 90 and 150 s for the automated and manual chambers, respectively, and the first 20 seconds after chamber closure was discarded (the dead band)."

We have moved and rearranged the second sentence, concerning the fan experiment, to section 2.3, page 5, line 20-25. It now reads:
"A dead band of 60 s, longer than the 20 s used for the one year campaign, was required, as well as a longer closure time, because an external gas analyser was attached to the LI-8100A during the fan experiment. The longer chamber closure time was used because the external analyser required a larger difference in $CO_2$

concentration during chamber measurements to achieve sufficient precision. The longer dead band was used because the extra volume added to the system by the external gas analyser caused longer response times and therefore also longer time to achieve stability after chamber closure."

**Reply to reviewer 3**

We have made the following changes to the manuscript in accordance with the reviewer's suggestions:

*"Major comments: 1. Page 3 line 32-33 – Maybe I understood wrongly, but from my point of view the hypotheses is stated wrongly or not precisely enough. There is written that overestimation of the CO2 fluxes during stable atmospheric conditions was due to insufficient mixing of the air above the soil surface. – do you mind the air inside the chamber headspace or in the open air? This should be clarified. If chamber headspace is considered I would avoid such hypotheses as it is discussed already in several papers that the effect of overestimation of nighttime fluxes is due to broken down the highly stratified layer of air inside the chamber headspace due to chamber movement at closure (Gorres et al. 2015) or air mixing in the chamber headspace (Schneider et al. 2009, Juszczak et al. 2012), which lead to change of predeployment steady state steep CO2 concertation gradient above the soil due to air mixing in the chamber. However, if insufficient air mixing in the atmosphere is considered then I would avoid to promote any disruptions of this natural condition occurring during calm nights (by excessive artificial air mixing, as authors has suggested, as a possible solution to overcome the overestimation of night-time fluxes) as this again will artificially change the concentration gradients between the soil and the atmosphere and enhance emission of gases, which does not occur during nights with stable atmospheric conditions when the air is highly stratified and when the only process driving emission of CO2 from the soil is diffusion."*

We have paraphrased page 4, line 1-3 (originally page 3, line 32-33 in the first manuscript). We have clarified that we mean that overestimation of the $CO_2$ fluxes during stable atmospheric conditions is due to insufficient mixing of open air above the soil surface. The lines now read:
"The second aim was to test the hypothesis that the overestimation of soil $CO_2$ effluxes during low $u_*$ was due to insufficient mixing of the open air above the soil surface and to test if unbiased soil $CO_2$ efflux measurements could be achieved during low $u_*$ by artificially inducing mixing of the air around the soil chambers by a fan. "

*"2. Page 4 line 9 – could you please specify how dense the canopy is? This seems to be not so important for the paper but it helps to imagine how far a density of the forest canopy may impact the turbulences in the canopy, especially that the sonic anemometer was installed 43 meters above the soil surface. In the ecosystems with a short vegetation (e.g. grasslands), the u\* filtering procedure can be applied to separate periods with calm and turbulent atmospheric conditions near the surface, but in a forest canopy with nearly 30 meters height of tree stands this might be more difficult, as under certain conditions the air might not be well mixed under canopy, although there are turbulences in the air above the canopy. Can you be sure that during turbulent conditions (at height of 43 m) there are still turbulences near the soil surface? Question is how dense the forest is? this will help to interpret the data you have. Maybe, it would help if you look for CO2 storage and relate this amount to measured FCO2 from EC? If storage is relatively high the air is not well mixed and may be stratified in the forest canopy"*

We have added the following information on page 4, line 16-17 regarding canopy density:

"The average annual duration of canopy cover is 180 days with a peak LAI of 5.0, and the tree stem density is 266 per hectare."

*"3. Page 5 line 25-30 regarding closure time (line5 of page 5) there is written that closure time was 90 and 150 sec in automated and manual chambers respectively, while for a fan experiment you extended the closure time to 5 minutes. Why the closure time was different? In case of manual measurements, the first 20 sec of data points were discarded (due to initial disturbances), while in case of a fan campaign you discarded first 60 seconds: : : this was due to time-leg the air came from the chamber to the analyzer? What was the length of tubes ? Was it the same for all chambers? Regarding fluxes please specify how the fluxes were calculated for automated chambers. Did you calculate fluxes with your R script or you relied on fluxes calculated by LICOR soft? Were the same quality criteria taken into account for all chambers (page 5 line 28- 30)?. If not, is that mean that fluxes which does not pass the goodness-of-fit criteria were also taken for analyses? (in manual chambers number of fluxes is small hence the question is what kind of quality criteria were applied in this case). Another point is that the linear fitting was applied to calculate fluxes. This is absolutely correct if closure time is short, but in the case of a fan experiment your closure time was 2-3 times longer than in manual and automated chambers – as described above. I assume that if you used the same small/short LICOR chambers there is for sure non-linear development of $CO_2$ concertation in the chamber headspace. In this case we know that if linear fitting is applied to calculate fluxes then this will lead to significant flux underestimation. Did you consider this effect?"*

We have added the following information regarding how the fluxes were calculated using R to page 6, line 5-9:
"However, non-linear effluxes were calculated as well, and at four different dead bands of 10, 20, 30 and 40 s (see supplementary information). The linear fluxes were calculated in R (R Core Team, 2014) using the lm function and the non-linear fluxes were calculated by fitting the non-linear equation suggested by Hutchinson and Mosier (1981) with the nlsLM function (minpack.lm package) for model fitting in R."

We have added information regarding quality control of the manual measurements on page 6, line 13-16. It reads the following:
"For the manual measurements, the quality control was done in the field directly following a measurement. If the coefficient of variance of the flux (as provided by the LI-COR software immediately following a measurement) was higher than 1.4, the measurement was discarded and an extra measurement was performed on the soil collar."

*"Considering above, please clarify how the annual fluxes were calculated. If the measured fluxes were divided to 12 different subsets (depending on u*), then for sure you had different numbers of fluxes for each class (please specify this information e.g. in Fig. 4). Please specify how the annual fluxes were calculated then? I understood that first data were filtered based on u* and 12 subsets were selected. That means that you had for each day different number of fluxes for each chamber – these might be daytime and night-time fluxes or only daytime or only night time fluxesˇAˇThow the daily fluxes were calculated then? Here, I would suggest to look for Rs vs T relationship for each subset and use e.g. Lloyed and Taylor function to model Rs for the whole year. If it is done in other way, I consider this incorrect, especially that we do not know how many fluxes were in each group and from which part of the day. This may impact results described in section 3.2 and 3.3., the whole discussion of results and might be critical point for the analyses presented in the paper"*

We have included information regarding the number of soil $CO_2$ effluxes in each of 12 different sub-datasets each with a specific $u_*$ threshold on page 6, line 22-24. It reads the following:

"The number of soil $CO_2$ effluxes was 43505 for the unfiltered dataset. For the 12 different sub-datasets going from a $u_*$ threshold value of 0.1 to 1.2 m s$^{-1}$, the number of soil $CO_2$ effluxes was 32966, 26848, 22185, 18557, 15787, 13449, 11472, 9533, 7950, 6571, 5481 and 4571, respectively. "

*"5. From data you presented is clear that CO2 effluxes are not following soil temperature changes over the day and they are inversely dependent on u\*. Of course there is a weak diurnal change of soil temperature at 5 cm depth and mainly it appears during summer, but still I would expect higher fluxes in the afternoon when soil temperature reaches maximum. The filtering procedure you applied lead to significant reduction of the fluxes during nights and slight reduction of daytime fluxes (which also indicates that there were stable atmospheric conditions over the day, as also was indicated by a fan experiment), hence still Rs was the smallest in the afternoon (besides autumn Fig 5j). Considering above I am wondering whether the temperature sensors are installed correctly? Maybe they are too deep? Have you measured soil temperature at 2 cm depth? Is there any relationship between air temperature (near the surface) and soil temperature? If not, then I assume the trees canopy might be so dense that the soil surface is homogenously shadowed, but this may also mean that next to the surface (where chamber measurements are conducted) there might be not much turbulences (this should also be critically discussed in the paper). Please specify in methods how many temperature sensors were installed and how they were distributed over the site – were they installed in soil collars, next to, or few meters from? – this information is missing although it may help to understand why there is no correlation between measured fluxes and T (on daily basis). I assume this might be autotrophic respiration of tree roots which may dominate your Rs and may not be depend on temperature and if yes, the data analyses would be even more complicated."*

We have added the following to page 5, line 11-13 regarding the number of temperature sensors and how they were distributed for the automated measurements:
"For the automated measurements, 6 soil thermometers were distributed close to the soil chambers, such that no soil chamber was further away than 10 m from an individual soil probe."

We have also specified the number of soil thermometers in the Fig. 3 caption. It now reads:
"Figure 3: Seasonally averaged diurnal pattern of soil temperature (± standard deviation) at 5 cm depth measured by 6 soil thermometers close to the eight automated chambers for summer (a), autumn (b), winter (c) and spring (d)."

*"Other minor comments and suggestions:*
*page 2 Line 18 – there was also a paper of Pumpanen et al. (2004) where rates of over- or underestimation of CO2 fluxes measured by different chamebers are presented in the controlled conditions, it is worth to cite it here"*

We have cited Pumpanen et al. (2004) on page 2, Line 18.

*"page 3 line 14-15, I would not agree that the mechanisms leading for flux overestimation are uncertain. They are well discussed in the cited papers and also in your paper (page 9-10) hence I would remove this sentence."*

We have deleted the sentence "The mechanism leading to flux overestimation, however, remains uncertain." on page 3, line 16.

*"Page 3, line 12 – please consider also whether to cite the paper of Juszczak et al. (2012) in Polish Journal of Environmental Studies, who compared daytime and nighttime Reco fluxes measured manually by chambers and filtered fluxes based on u\*, and proved that when proper flux filtering (based on u\*) is applied then there is no difference between day and night-time REco fluxes measured by chambers, while they are significantly overestimated when no filtering is applied. This is in agreement with your statements (page 11, lines 19-24)."*

We have cited the Juszczak et al. (2012) on page 3, line 13.
We have also referred to the paper following the discussion on page 12, line 17-21. It reads the following on page 12, line 21-23:
"This is in agreement with Juszczak et al. (2012) that found that there was no difference between day-time and night-time effluxes when the correct $u_*$ threshold value had been applied."

*"Page 5 line 15-20 please specify the height the fans are installed. The chamber you used are rather small/short and I assume fans were just above the soil surface? But this need to be written here"*

We have included information regarding fan height on page 5, line 27-29. The sentence now reads:
"The artificial air mixing for each chamber was provided by 30 cm diameter table fans facing the chamber (Model 546601, HP Schou A/S, Kolding, Denmark) positioned 3 m from the soil chamber and at a height of 30 cm from the soil surface to the middle of the fan."

*"Page 5 line 10-20 Can you please clearly write in paragraph 2.3 what kind of chambers were used and why you extend the closure time in case of a fan experiment."*

We have specified which chambers were used and why the extended closure time was used in section 2.3. Page 5, line 18-25 now reads:
"During a 20 day campaign in July and August 2016 soil $CO_2$ effluxes were measured at the site with four of the 8100-104 Long-Term $CO_2$ flux chambers and two of the 8100-101 Long-Term $CO_2$ flux chambers, with each chamber measuring soil $CO_2$ effluxes every two hours using a chamber closure time of 5 minutes. A dead band of 60 s, longer than the 20 s used for the one year campaign, was required, as well as a longer closure time, because an external gas analyser was attached to the LI-8100A during the fan experiment. The longer chamber closure time was used because the external analyser required a larger difference in $CO_2$ concentration during chamber measurements to achieve sufficient precision. The longer dead band was used because the extra volume added to the system by the external gas analyzer caused longer response times and therefore also longer time to achieve stability after chamber closure."

*"Page 5 line 31, write covariance instead of co-variance"*

We have changed "co-variance" to "covariance" on page 6, line 17.

*"Page 12 lines 34 page 13 lines 1-5 – this was already suggested if I well remember in Rochette and Hutchinson (2005),. They suggested that to avoid overmixing of the air in the chamber headspace the fan speed should be adjusted to outside wind speed."*

We have included a reference to Rochette and Hutchinson (2005). Page 13, line 34 to page 14, line 2 now reads:

"To eliminate this potential bias due to difference in wind speed, a closer matching of the chamber wind speed with the ambient wind speed can be attempted, as suggested by Rochette and Hutchinson (2005)."

**Reply to Thomas Foken**

We have made the following changes to the manuscript following Thomas Foken's suggestions:

*"It is a pity that no additional sonic anemometer was installed in the trunk space, which was recently urgently recommended (Thomas et al., 2013). Therefore you are unable to indicate if the atmosphere above the canopy (where you measured the friction velocity) is coupled with the trunk space (Thomas and Foken, 2007). Your daily and annual cycle of the friction velocity is probably slightly modified by the significant daily and annual cycle of coupling (Foken et al., 2012; Jocher et al., 2017). Perhaps you should include in your recommendations a second sonic anemometer, which would control the turbulent mixing (friction velocity, standard deviation of the vertical wind velocity) in the vicinity of the soil chambers. This may reduce the proposed very high u\*-threshold."*

We have added the following regarding anemometer height in section 4.5, page 14, line 31 to page 15, line 3:
"One possible explanation is that the $u_*$ used in this experiment was measured above the canopy at a height of 43 m. The soil chambers, however, were positioned at the soil surface and it is possible that the measured turbulence at 43 m was not well coupled to $u_*$ above the soil surface surrounding the soil chambers (Thomas and Foken, 2007). A second anemometer installed closer to the soil surface (see Thomas et al., 2013) might have provided a clearer $u_*$ threshold value."

*"The problem you addressed is not only related to the turbulent mixing or the friction velocity. Low friction velocities are often connected with stable stratification and the reason for very stable conditions near the surface is a cooling by a large longwave net radiation. In contrast to the natural condition, the longwave net radiation in a chamber is always nearly zero and therefore the stratification is always nearly neutral. This may also be a reason why a chamber under stable (night-time) conditions can overestimate the fluxes (Riederer et al., 2014). Our study was made above a meadow with much larger longwave net radiation than inside the canopy, but nevertheless the longwave radiation effect on chamber measurements should be discussed. Helpful would be four-component net radiometers above the forest and in the trunk space – perhaps a further recommendation for flux sites."*

We have added the following regarding stable conditions due to longwave net radiation in section 4.1, page 11, line 1-4:

[revised manuscript text omitted]